# Predatory synapsid ecomorphology signals growing dynamism of late Palaeozoic terrestrial ecosystems
Suresh A. Singh [1] ✉, Armin Elsler [1], Thomas L. Stubbs [2], Emily J. Rayfield [1] & Michael J. Benton [1]

Terrestrial ecosystems evolved substantially through the Palaeozoic, especially the Permian, gaining much new complexity, especially among predators. Key among these predators were non-mammalian synapsids. Predator ecomorphology reflect interactions with prey and competitors, which are key controls on carnivore diversity and ecology. Therefore, carnivorous synapsids may offer insight on wider ecological evolution as the first complex, tetrapod-dominated, terrestrial ecosystems formed through the late Palaeozoic. Using morphometric and phylogenetic comparative methods, we chart carnivorous synapsid trophic morphology from the latest Carboniferous to the earliest Triassic (307-251.2 Ma). We find a major morphofunctional shift in synapsid carnivory between the early and middle Permian, via the addition of new feeding modes increasingly specialised for greater biting power or speed that captures the growing antagonism and dynamism of terrestrial tetrapod predator-prey interactions. The further evolution of new hypo- and hypercarnivorous synapsids highlight the nascent intrinsic pressures and complexification of terrestrial ecosystems across the mid-late Permian.

Tetrapod terrestrialisation through the late Palaeozoic is a pivotal moment in Earth history, as tetrapods revolutionised life on land by adding new complexity to terrestrial trophic networks, establishing the basic relationships that still underpin terrestrial ecosystems today[1–5]. Overcoming multiple organismal and environmental constraints, tetrapods became increasingly adept on land as they evolved to better survive and exploit the resources of their new realm[6–10]. By the late Permian, this diversification produced rich communities of specialist tetrapod herbivores and carnivores, echoing the diversity of modern ecosystems[11]. Nonetheless, Palaeozoic ecosystems often differed structurally from modern counterparts by possessing disproportionately diverse carnivore contingents[12–14]. Such predator-rich terrestrial faunas appeared throughout the Palaeozoic and Mesozoic[12,13,15,16], in contrast to more prey-rich systems that dominated the Cenozoic[3,17]. These differences raise the possibility of substantially differing ecological dynamics through deep time and point to the need for detailed understanding of such ancient ecosystems and their influence on tetrapod macroevolution.

A limited fossil record precludes direct analysis of Palaeozoic ecological interactions and processes[18], but such interactions are a key selective pressure in evolution, driving behavioural shifts that ultimately promote phenotypic change[19–21]. Therefore, functional anatomy may provide a window onto these interactions through deep time. Carnivores, especially large macropredators, are often useful indicators of ecological change[22,23] as they exert great influence over their ecosystems through antagonistic relationships with prey and competitors, which in turn, are major influences on carnivore behavioural ecology, forcing changes in habitat, diet, and foraging activity[3,24,25]. Palaeozoic terrestrial ecological evolution may therefore be studied using the ecomorphology of the leading terrestrial carnivores of the time: non-mammalian synapsids.

Synapsids rose quickly to prominence within the terrestrial carnivore guild during the first major radiation of terrestrial amniotes in the Late Carboniferous[26], with basal, 'pelycosaur-grade' synapsids becoming the top terrestrial predators by the early Permian[27,28]. Despite extinction events at the end of the early and middle Permian that eliminated much of their diversity, synapsids maintained a monopoly on large terrestrial carnivore niches to the end of the Palaeozoic, with successive diversifications of therapsids creating rich, new carnivore communities, dominated by biarmosuchians and dinocephalians in the Guadalupian, and then gorgonopsians and therocephalians in the Lopingian[2,29]. Synapsid faunal dominance was ended by the Permo-Triassic Mass Extinction (PTME), allowing diapsid archosauromorphs to overtake them through the Triassic[4]. Synapsid monopolisation of the terrestrial carnivore guild

¹School of Earth Sciences, University of Bristol, Life Sciences Building, Tyndall Avenue, Bristol BS8 1TQ, UK. ²School of Life, Health and Chemical Sciences, Open University, Milton Keynes MK7 6AE, UK. ✉e-mail: sureshsingh.palaeo@gmail.com

through the late Palaeozoic offers an opportunity to study trophic ecological dynamics through the founding and development of the first complex tetrapod ecosystems on land, as well as multiple mass extinction events[5,30].

By applying morphometric and macroevolutionary analytical methods including the new consensus clustering method of Singh et al.[31], we reconstruct and quantitively classify the feeding ecologies of carnivorous non-mammalian synapsids through the Late Carboniferous—Early Triassic (315.2–251.2 Ma), using jaw functional morphology and body size, both of which closely relate to feeding and foraging behaviour[32–35]. Through a combination of geometric and traditional linear measurement-based morphometric methods[31], we provide a broad assessment that captures synapsid jaw morphofunctional evolution from differing perspectives and partially mitigates the divergent impacts of phylogenetic heritage, taxonomic scaling, or methodological choices[36]. Even though non-mammalian synapsid jaw functionality uniquely encompasses a spectrum between reptiles and mammals unseen in extant taxa[37], changes in basic functional properties (e.g., mechanical advantage or symphyseal robusticity) allow us to use absolute and relative similarities to living animals to make some rudimentary inferences and hypotheses of non-mammalian synapsid prey preferences, modes of prey capture, and consumption that can be further examined in more bespoke, future biomechanical studies. Through the identification and comparison of the distinct functional feeding groups (FFGs) of carnivorous synapsids, we extract new details on the trophic interactions of terrestrial tetrapods, revealing their ecological evolution through the Palaeozoic.

## Results and discussion

### Synapsid carnivore jaw morphofunctional diversity

Geometric morphometric landmark data (Supplementary Fig. 1 and Supplementary Data 1) and standardised functional measurements (SFM) (Supplementary Fig. 2 and Supplementary Data 2) were used to study synapsid jaw evolution following the approach of Singh et al.[31], as they provide two slightly different but complementary perspectives on jaw morphofunction[36] (see Supplementary Methods). Both approaches are tied to morphology, but the geometric method captures unambiguous differences in shape, whereas the functional measurement approach provides more direct assessment of potential functionality across different jaw morphologies. This approach also provides some assurance that trends and differences shown here are grounded in real patterns of change as non-mammalian synapsid jaw evolution encompasses significant changes in jaw structure, musculature and function[37]. We henceforth interpret and refer to these analyses as more reflective of jaw form and function, labelling them as the 'shape' and 'function' analyses, respectively. Furthermore, trends in form and functional evolution are often linked, but do not necessarily correspond[36].

The primary axes of jaw form and functional variation are illustrated using principal component analyses (PCA) and morphospaces constructed from the resulting first two principal components (PCs), which represent 40.7% of total shape variation and 60.6% of total functional variation (Supplementary Tables 1 and 2). Jaw form varies most significantly along PC1 through the changing depth of the symphysis and mandibular body, and the curvature of the mandibular ramus, whereas PC2 marks the relative prominence of a coronoid process (Fig. 1a). Functional trait mapping across

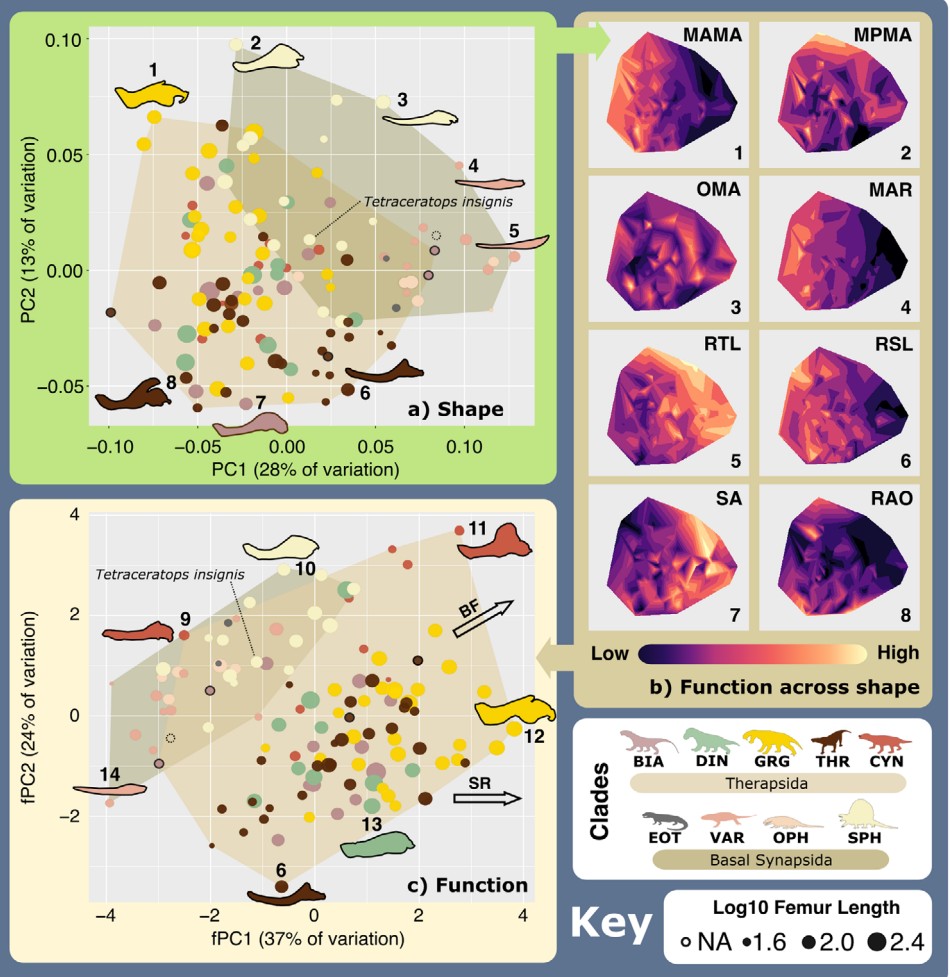

**Fig. 1 | Synapsid carnivore jaw morpho-functional diversity. a** Jaw shape morphospace. **b** Jaw functional characters mapped across shape morphospace. (Colour gradient reflects functional character values—see scale.) **c** Jaw functional morphospace, with arrows showing general functional trends. Point size represents Log$_{10}$(mm) femur length. $N = 122$. Jaw silhouettes: 1. *Smilesaurus ferox*, 2. *Sphenacodon ferox*, 3. *Secodontosaurus obtusidens*, 4. *Microvaranops parentis*, 5. *Varanodon agilis*, 6. *Lycideops longiceps*, 7. *Lobalopex mordax*, 8. *Ictidosaurus angusticeps*, 9. *Procynosuchus delaharpeae*, 10. *Dimetrodon milleri*, 11. *Vetusodon elikhulu*, 12. *Dinogorgon rubidgei*, 13. *Deuterosaurus biarmicus*, 14. *Mycterosaurus longiceps*. BF biting force, BIA Biarmosuchia, CYN Cynodontia, DIN Dinocephalia, EOT Eothyrididae, GRG Gorgonopsia, MAMA mean anterior mechanical advantage, MAR maximum aspect ratio, MPMA mean posterior mechanical advantage, OMA opening mechanical advantage, OPH Ophiacodontidae, RAO relative articulation offset, RSL relative symphyseal length, RTL relative toothrow length, SA Symphyseal angle, SPH Sphenacodontia (non-therapsid), SR Symphyseal robusticity, THR Therocephalia, VAR Varanopidae. $N = 122$ taxa. Biarmosuchia, Dinocephalia and Therocephalia silhouettes by Dmitry Bogdanov (vectorized by T. Michael Keesey); all other silhouettes created by S.A.S., but some are vectorised from artwork by Felipe Alves Elias (https://www.paleozoobr.com/), available for academic use with attribution.

the shape morphospace (Fig. 1b) reveals marked functional heterogeneity across jaw shape, particularly in mean posterior mechanical advantage (MA), opening MA, and symphyseal angle. Some patterns exist as PC1 negatively relates to mean anterior mechanical advantage, maximum aspect ratio, and relative symphyseal length, but positively with relative toothrow length. The functional morphospace (Fig. 1c) generated from the SFMs (Fig. 1b) shows that taxa are principally distinguished by mean anterior MA, maximum aspect ratio, and relative toothrow length along functional PC (fPC) 1 (Supplementary Table 3). Anterior and posterior MA are the respective prime determinants of fPC1 and fPC2, but fPC2 shows additional strong relationships with relative toothrow length and opening MA. Consideration of body size represented using $\log_{10}$ femur length (Supplementary Data 3) shows that jaw robusticity and biting efficiency scale positively with size.

Both the form and functional morphospaces show parallel distributions in both basal synapsids and therapsids from relatively gracile, elongate jaws towards more robust morphologies capable of more powerful bites (Fig. 1). Basal synapsids and therapsids are distinguished principally by reductions in relative toothrow length and prominence of the postdentary bones in therapsids[29]. Both groups occupy similar extents of shape morphospace despite differences in sampling, but therapsid functional morphospace exceeds that of basal synapsids. Additional morphospaces constructed using PC/fPC3 (9.3% and 11.4% of shape and functional variation, respectively), also show basal synapsids and therapsids distributed broadly in parallel across shape morphospace (Supplementary Fig. 3a), and greater therapsid functional morphospace occupation (MO) (Supplementary Fig. 3b). PC3 captures the relative size and curvature of the jaw, most distinctively in the surangular, whereas fPC3 generally represents the curvature of the ramus.

Subclade MO highlights strong trends in jaw form and function through synapsid evolution. Basal synapsids developed increasing robusticity and enlargement of the mandibular body from varanopids and ophiacodonts to (non-therapsid) sphenacodontians (Fig. 1 and Supplementary Fig. 4). This pattern extends to therapsids, as taxa within multiple clades (particularly gorgonopsians) evolved more robust jaw morphologies with reinforced symphyses, although some taxa contrastingly evolved highly gracile morphologies with curved mandibular rami (Fig. 1). Gorgonopsians and therocephalians are generally quite similar but show some differences in functional character ranges that indicate divergent optimisations for power or speed (Supplementary Fig. 4). Cynodont MO intriguingly extends across basal synapsid and therapsid MO with optimisation of posterior biting efficiency and relatively large toothrows (Fig. 1 and Supplementary Fig. 4). PERMANOVA reveals significant differences in jaw form and function between most synapsid groups, highlighting their disparity (Supplementary Tables 4 and 5).

## Synapsid carnivore functional feeding groups

Active carnivores apply a mix of compressive, shearing, tearing, and puncture damage, and each aspect acts differently to incapacitate prey[38]. Puncture and compressive injuries extend damage deeper within prey tissue, potentially reaching vital internal organs, whereas shearing and tearing are focused on inflicting trauma through tissue and blood loss[39,40]. Different prey capture behaviours inflict different combinations of damage, which are likely reflected in carnivore jaw morphofunction. Using a consensus cluster method[31], we quantitatively identify three functional feeding groups (FFGs) from the SFMs, characterised as raptorial specialists, power shearers, and speed specialists (Supplementary Fig. 5). Further consensus cluster analysis of each FFG identified seven, more subtle and specific feeding functional subgroups (FFsGs) (Fig. 2; Supplementary Fig. 6a and Supplementary Data 4). FF(s)G classifications were validated using a jack-knifed, linear discriminant analysis (LDA)[41] and show coherent distributions across the form and function morphospaces (Supplementary Fig. 6). Despite some phylogenetic sorting that nonetheless reflects hard eco-functional differences, external validation metrics show low correspondence between cluster classifications and phylogeny at higher taxonomic levels (Supplementary

Tables 6 and 7). Each FF(s)G shows clear differences (Supplementary Tables 8–10) and further assessment of their functionalities highlight their particular feeding ecologies:

- **Raptorial specialists:** This group is united by their gracile, long-irostrine jaws and lengthy toothrows, and subdivided by differences in robusticity and biting efficiency into the 'gracile and forceful grippers' (GG and FG) subgroups (Fig. 2 and Supplementary Figs. 5 and 6). This group is almost exclusively populated by basal synapsids but includes some biarmosuchian therapsids. Varanopids and ophiacodonts comprise the majority of GG, and larger robust members of both clades as well as most sphenacodontians form the FG subgroup (Fig. 2 and Supplementary Fig. 7). Extended toothrows enable a wide distribution of bite force and suggests an emphasis on gripping and retaining prey, particularly when combined with the conidont teeth present in basal synapsids (Figs. 2 and 3 and Supplementary Fig. 6). Their gracile jaws (particularly of the GG) are ill-suited to high stresses associated with comparatively large prey (Fig. 2), suggesting a preference for much smaller, less combative prey such as insects, fish, and smaller tetrapods (Fig. 3). Differences between GG and FG raptorial specialists in MA, areas of muscle attachment, and dentary robusticity (Figs. 1b and 4 and Supplementary Fig. 6) (Fig. 2) illustrate FG optimisation for biting efficiency and power over biting speed[28]. Growing tooth size and shape variation through basal synapsid evolution[42] supports a shift towards more complex jaw use and feeding behaviour associated with tetrapod-on-tetrapod predation. Raptorial specialist dentition encompasses simplistic conidont to derived ziphodont tooth morphologies, with the ziphodont teeth appearing more commonly in sphenacodontids[42,43], illustrating their growing efficiency for shearing flesh and specialisation as tetrapod predators[44]. Raptorial specialists somewhat echo the jaw functionality of some sauropsid reptiles, and such similarity may extend to prey capture/killing behaviour (Fig. 3). Varanopids were well suited to rapid head movements[45] like modern varanid lizards, which employ such movements when grasping and killing prey[37,46]. The basal phylogenetic position of varanopids indicates such behaviour may be plesiomorphic for synapsid predators. However, varanid prey capture and dismemberment heavily involve their neck and forelimbs[47,48], whereas basal synapsids likely relied primarily on their jaws to manipulate prey.

- **Speed specialists:** All therapsid clades, particularly therocephalians, are represented within this group, which is subdivided into 'grip and rip attackers' (GRA) and 'rapid light attackers' (RLA) (Fig. 2 and Supplementary Fig. 6). Much like the raptorial specialists, speed specialists exhibit low MA indicative of fast bite speeds[33], moderate robusticity, and prominently curved dentaries to improve their grip on prey (Fig. 2 and Supplementary Figs. 5 and 6). RLA speed specialists modified these traits further, in addition to reducing their OMA to enhance bite speed (Fig. 2). However, speed specialists show more limited distribution of biting force towards the front of the jaw, as typically illustrated by a shorter toothrow and reinforced symphysis. Such modifications demonstrate enhanced focus on the penetrative and gripping power of the jaws. RTL is generally shorter than in raptorial specialists but therapsid speed specialists still possess lengthier toothrows than seen in therapsid power shearers (Figs. 1b and 2 and Supplementary Fig. 6). These toothrows generally feature highly enlarged 'pre-canine' and 'canine' teeth, and smaller but often more complex post-canine teeth[42]. Greater emphasis on anterior biting efficiency boosted the penetrative power of the anterior dentition, while greater post-canine complexity combined with increased jaw robusticity further enhanced the ability to hold and resist struggling prey. Alongside higher biting speeds, these features suggest that these therapsids potentially employed a 'harrying' manner of prey capture, like that seen in canids[49] (Fig. 3). Such behaviour is somewhat consistent with jaw and neck muscle development across the basal synapsid-therapsid transition, although therapsids lacked the axial flexibility and speed[50] to fully replicate canid hunting, which often involves extended pursuits[49]. While better suited

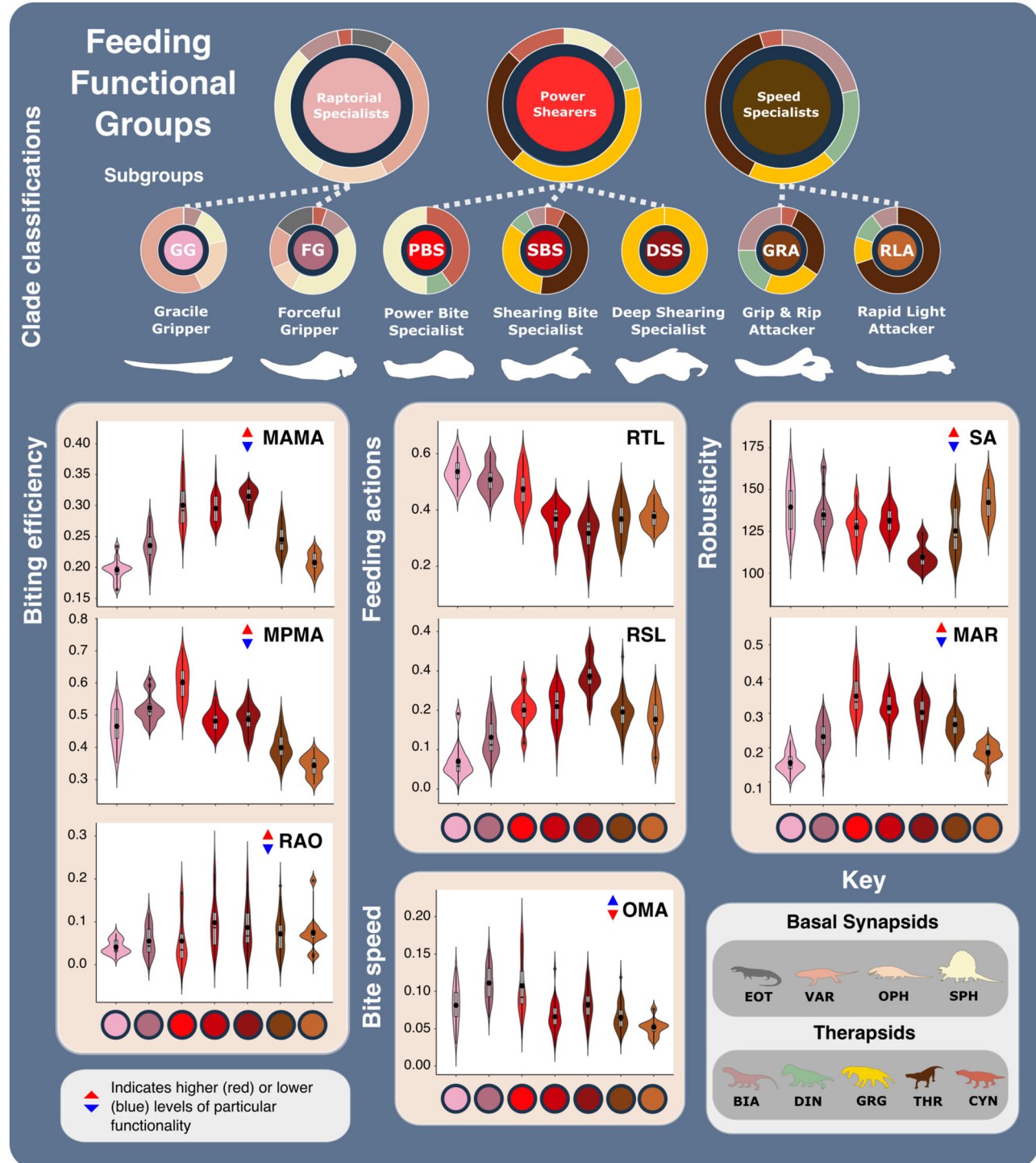

**Fig. 2 | Synapsid carnivore feeding functional subgroup jaw characteristics.** The feeding functional subgroup jaw functional character (Supplementary Methods) distributions illustrated using violin and box plots. Functional feeding group compositions illustrated using ring plots detailing relative proportions of different taxonomic groups. Violin plots show taxon density. Box plots showing median value and upper and lower quartiles, with whisker illustrating standard deviation. Mean values indicate by black dots. Coloured arrows indicate whether values increase (red) or decrease (blue) relevant jaw functionality. N = 122. Jaw silhouettes (left to right): *Varanodon agilis*, *Tetraceratops insignis*, *Dimetrodon grandis*, *Sauroctonus parringtoni*, *Smilesaurus ferox*, *Annatherapsidus petri*, *Tetracynodon darti*. BIA

Biarmosuchia, CYN Cynodontia, DIN Dinocephalia, EOT Eothyrididae, GRG Gorgonopsia, MAMA mean anterior mechanical advantage, MAR maximum aspect ratio, MPMA mean posterior mechanical advantage, OMA opening mechanical advantage, OPH Ophiacodontidae, RAO relative articulation offset, RSL relative symphyseal length, RTL relative toothrow length, SA Symphyseal angle, SPH Sphenacodontia (non-therapsid), THR Therocephalia, VAR Varanopidae. N = 122 taxa. Biarmosuchia, Dinocephalia and Therocephalia silhouettes by Dmitry Bogdanov (vectorized by T. Michael Keesey); all other silhouettes created by S.A.S., but some are vectorised from artwork by Felipe Alves Elias (https://www.paleozoobr.com/), available for academic use with attribution.

**Fig. 3 | The ecofunctional focus of synapsid carnivore functional feeding groups.** Likely prey preferences and capture methods of the raptorial specialist, speed specialist, power shearer functional feeding groups, as suggested by overall interpretation of jaw functional traits. Jaw silhouettes (left to right): *Mesenosaurus romeri*, *Annatherapsidus petri*, *Aelurognathus tigriceps*. DSS deep shearing specialist, FG forceful gripper, GG gracile gripper, GRA grip and rip attacker, PBS power bite specialist, RLA rapid light attacker, SBS shearing bite specialist, SS speed specialist. All silhouettes created by S.A.S.

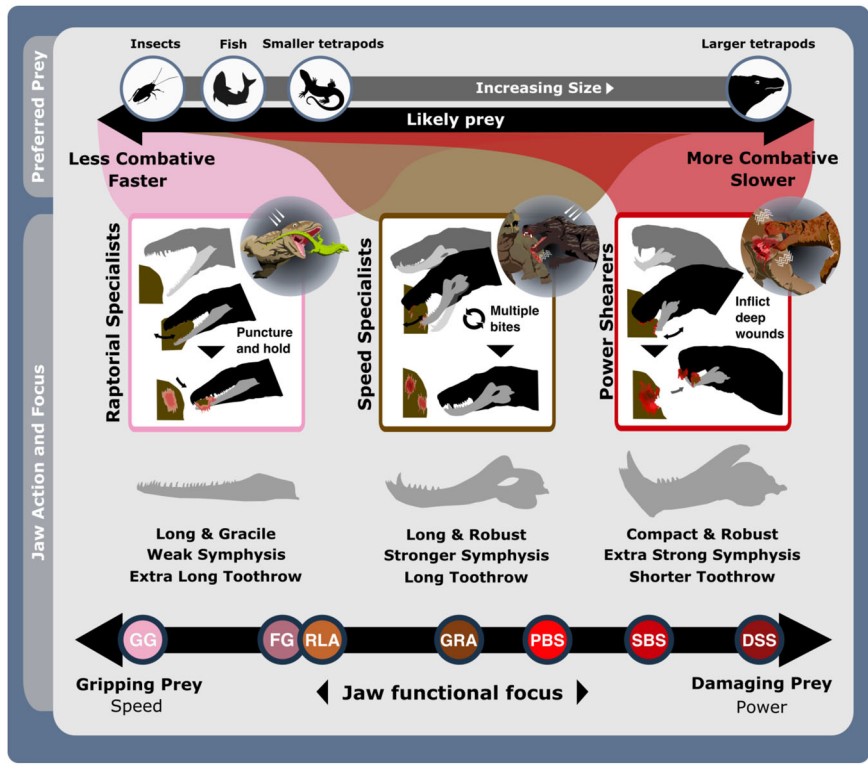

than the raptorial specialists to extended struggles with prey, the elongate jaws of GRAs remain unsuited to the higher stresses associated with such struggles with large prey. Therefore, GRAs likely targeted relatively smaller prey; using their own larger size to inflict disproportionately more damage and quickly subdue prey[51] (Figs. 2 and 3 and Supplementary Fig. 7). More robust forms could perhaps tackle larger, more similarly sized prey.

- **Power shearers:** This group mostly comprises gorgonopsians and basal therocephalians, but also features most cynodonts and multiple species of *Dimetrodon* and *Sphenacodon* within the power bite specialist (PBS) subgroup (Figs. 2 and 3 and Supplementary Figs. 5 and 6a). This group is distinguished by highly robust and mechanically efficient jaws with substantial symphyseal reinforcement. While shearing bite specialists (SBS) form the core of the group, power bite specialists (PBS) and deep shearing specialists (DSS) represent distinct variations on the power shearer feeding mode, opting to maximise biting efficiency along the entirety or just the anterior margin of the toothrow, respectively (Fig. 2 and Supplementary Fig. 6). The DSS are dominated by rubidgeine gorgonopsids, which were generally the largest and most heavily built gorgonopsians[52]. SBS and DSS appear optimised for penetrating deeply into and shearing off chunks of prey tissue, supporting previous inferences of hypercarnivorous behaviour derived from highly incisiform anterior dentition[42,53] and cranial morphology[53,54]. Hypercarnivory is also evident from the prevalence of hypertrophied canines across gorgonopsians and therocephalians[37], and the symphyseal reinforcement exhibited across this FFG likely also evolved to support these teeth against lateral stresses as in sabertoothed cats[55]. In contrast, the PBS exhibit greater mean posterior MA, suggesting additional emphasis on compressive forces during feeding (Figs. 2 and 3). Nonetheless, differences in size (Supplementary Figs. 6 and 7), dentition[42] and musculature[29] within the PBS point to divergent ecologies, with hypercarnivory in the sphenacodontids and dinocephalians, and mesocarnivory or omnivory in cynodonts. Sphenacodontid PBS possessed large ziphodont and/or conidont anterior teeth, and more bulbous posterior teeth[42] featuring subrounded to ovoid cross

sections that afforded high structural strength[36]. This differentiation indicates dual emphasis on removing flesh and inflicting compressive damage to incapacitate prey rather than acute durophagous feeding behaviour[43]. Relatively high posterior MA and jaw robusticity (Fig. 1b) make these sphenacodontids well suited to grappling with comparably sized prey[43,56,57]. Similar emphasis on posterior biting efficiency is also observed in the dinocephalian PBS, *Anteosaurus magnificus*[37,58] (Supplementary Data 2). The more complex multi-cusped teeth seen in cynodont PBS are associated with enhanced comminution[59] and suggestive of more durophagous diets.

FFG designations (Figs. 2 and 3) capture a shift in synapsid jaw mechanics from something similar to those of sauropsid reptiles towards an eco-functionality more like mammals[29,37]. Non-mammalian synapsids possessed a kinetic inertial biting system in which the jaw muscles were most active when the jaw was open, utilising the jaw's mass and velocity to maximise bite force during closure[37,60]. Therefore, symphyseal morphology and anterior biting efficiency unsurprisingly show the most variation of all functional characters (Fig. 1b). Palaeozoic synapsid jaw evolution was largely directed towards maximising anterior bite force by expanding their jaw musculature, forcing shifts in adductor muscle attachment sites and lines of action, particularly across the basal synapsid-therapsid transition[56,61]. This included the development of the coronoid process from an incipient feature in sphenacodontids to highly-posterodorsally extended processes in theriodont therapsids, as well as the reduction of the postdentary bones[61,62]. In the lower jaw, the toothrow became shorter and the symphysis was reinforced, thereby increasing anterior biting efficiency and resistance to torsional stresses (Fig. 1 and Supplementary Fig. 4). These changes amplified functional differences between the upper and lower jaws, creating a 'hammer and anvil' setup that maximised bite force in the downward movement of the upper jaws ('hammer'), while the lower jaws were reinforced to act a buttress against rostral stresses ('anvil')[56,63]. These modifications enabled more effective penetrative bites, which, combined with increasingly incisiform, heterodont dentitions[42], point to an expansion of jaw functionality in therapsids.

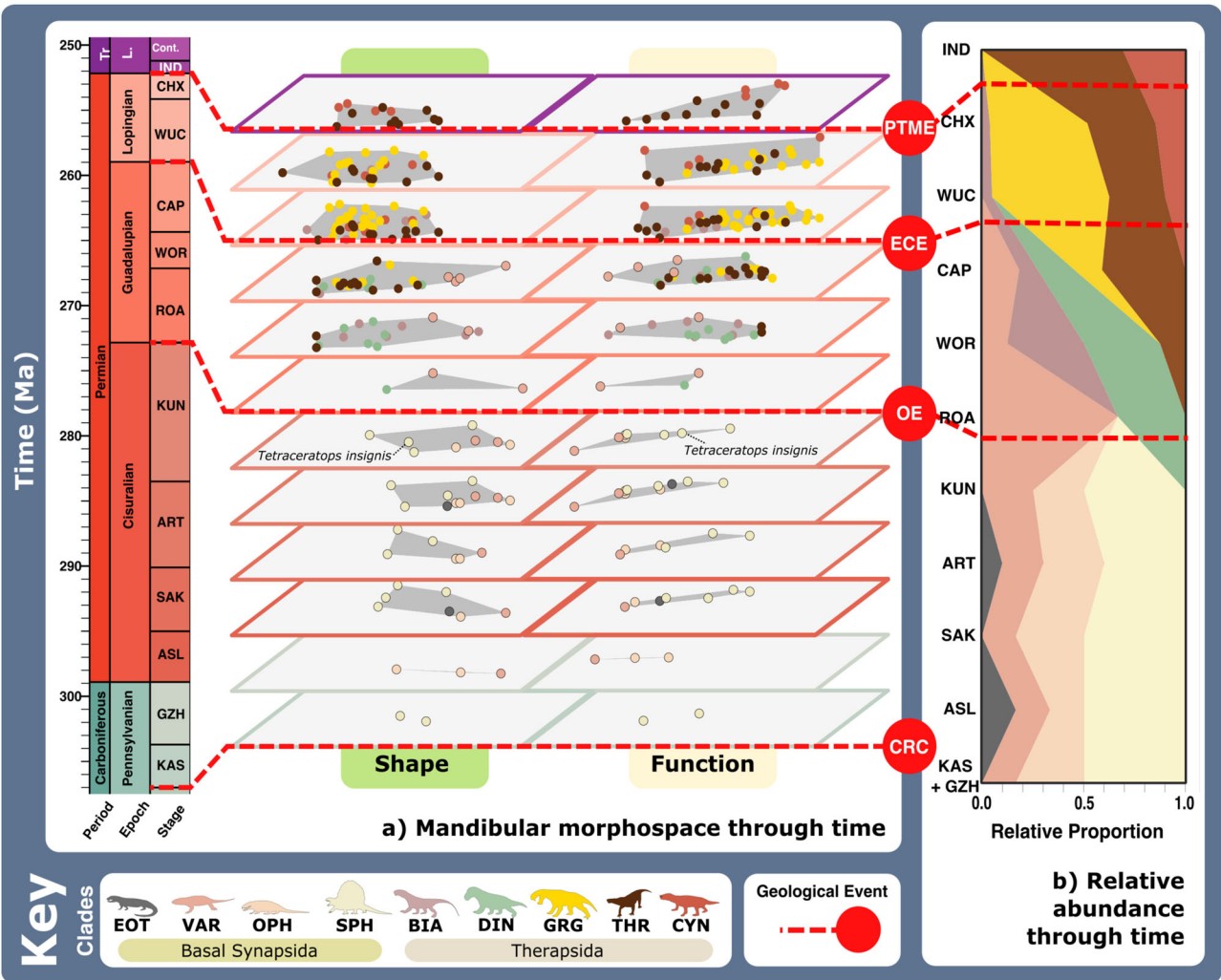

**Fig. 4 | Synapsid carnivore jaw morpho-functional evolution and relative abundance through time. a** Jaw shape and functional morphospace changes through the late Palaeozoic. Morphospace margin colours correspond to colours of the relevant time bin on the stratigraphic chart. **b** Relative proportions of different taxonomic groups per time bin through the late Palaeozoic. ART Artinskian, ASL Asselian, BIA Biarmosuchia, CAP Capitanian, CHX Changhsingian, CRC Carboniferous rainforest collapse, CYN Cynodontia, DIN Dinocephalia, ECE End-Capitanian extinction, EOT Eothyrididae, GRG Gorgonopsia, GZH Gzhelian, IND Induan, KAS Kasimovian, KUN Kungurian, OE Olson's extinction, OPH Ophiacodontidae, PENN Pennsylvanian, PTME Permo-Triassic mass extinction, SAK Sakmarian, SPH Sphenacodontia (non-therapsid), ROA Roadian, THR Therocephalia, VAR Varanopidae, WOR Wordian, WUC Wuchiapingian. Biarmosuchia, Dinocephalia and Therocephalia silhouettes by Dmitry Bogdanov (vectorized by T. Michael Keesey); all other silhouettes created by S.A.S., but some are vectorised from artwork by Felipe Alves Elias (https://www.paleozoobr.com/), available for academic use with attribution.

Evidently, the basal synapsid-therapsid transition saw an eco-functional shift from simply penetrating and holding prey, to also shearing prey tissue and thereby inflicting deeper wounds and heavier trauma[64]. Later Permian therapsid carnivores became specialised at heavily damaging and thus quickly incapacitating prey, making their prey capture methods more like mammalian rather than reptilian carnivores (Supplementary Fig. 4). The taxonomic composition and functional characteristics of our FFsGs (Fig. 2) align with this trend and support prior suggestions that greater adductive force and stabilisation when the jaws are open emerged as adaptations to counter the force of struggling prey[56,62]. Speed specialist jaw functionality suggests their jaws were deployed much like modern canids, but wider consideration of therapsid post-cranial anatomy suggests that therapsid predators more likely engaged in low-energy stalking, deploying a swift attack once within striking distance, more akin to felid carnivores[65]. Brevirostrine power-shearers share further similarities with felid predators by also emphasising biting power, likely using a few or perhaps one deep, slashing, debilitating bite aimed at fleshier areas of the body to maximise damage to critical anatomy and efficiently incapacitate prey[49,66,67] (Figs. 4 and 5).

The next major phase of synapsid jaw evolution was the reorganisation of the adductor musculature and jaw anatomy that created the mammalian jaw and middle ear[68]. While principally occurring in the Mesozoic, the inclusion of some cynodonts in the PBS (Fig. 2) may reflect the beginning of this transformation in the latest Permian. Repositioning the adductor attachment onto the dentary[61,69] marked a shift towards emphasising posterior biting efficiency (Fig. 1b) producing a static pressure jaw system in mammals, whereby force is principally exerted when the jaws are not in motion and/or almost closed[60].

### Synapsid carnivory through the late Palaeozoic

When carnivorous synapsid jaw morphofunctional disparity is viewed through time using stage-level plots of jaw shape and functional MO (Fig. 4 and Supplementary Data 5) and sum of variance curves generated from phylogenetic time-slicing[70] (Fig. 5), we find clear shifts in jaw morphofunction at the Carboniferous-Permian and early-middle Permian transitions. Further reconstructions of FFsG prevalence and overall body size across the carnivorous synapsid phylogeny (Figs. 6 and 7 and Supplementary Data 6–9) link functional morphology to wider

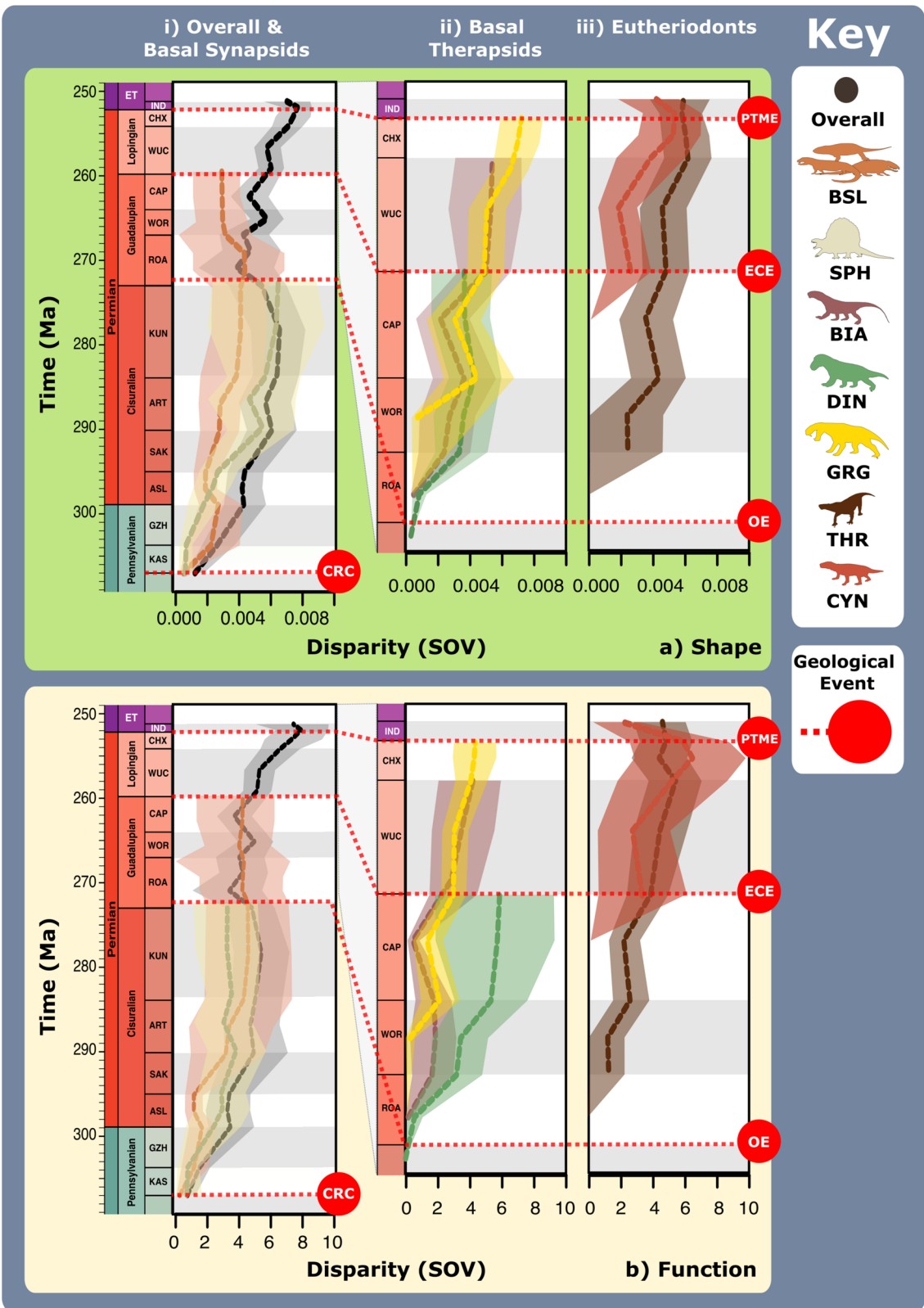

**Fig. 5 | Synapsid carnivore jaw shape and functional phylogenetic disparity through the late Palaeozoic. a** Shape and (**b**) functional sum of variance calculated for each time bin for carnivorous synapsid groups using phylogenetic time-slicing[70], divided into: (i) Basal synapsids, (ii) Basal therapsids, and (iii) Eutheriodonts. Significant geological events also highlighted. 'Overall' represents all carnivorous synapsids. Shaded 95% confidence intervals shown for each curve. $N = 122$. ART Artinskian, ASL Asselian, BIA Biarmosuchia, BSL Basal-most synapsids (eothyridids, varanopids, and ophiacodonts), CAP Capitanian, CHX Changhsingian, CRC Carboniferous rainforest collapse, CYN Cynodontia, DIN Dinocephalia, ECE End-Capitanian extinction, GRG Gorgonopsia, GZH Gzhelian, IND Induan, KAS Kasimovian, KUN Kungurian, OE Olson's extinction, PENN Pennsylvanian, PTME Permo-Triassic mass extinction, SAK Sakmarian, ROA Roadian, THR Therocephalia, WOR Wordian, WUC Wuchiapingian. Biarmosuchia, Dinocephalia and Therocephalia silhouettes by Dmitry Bogdanov (vectorized by T. Michael Keesey); all other silhouettes created by S.A.S., but some are vectorised from artwork by Felipe Alves Elias (https://www.paleozoobr.com/), available for academic use with attribution.

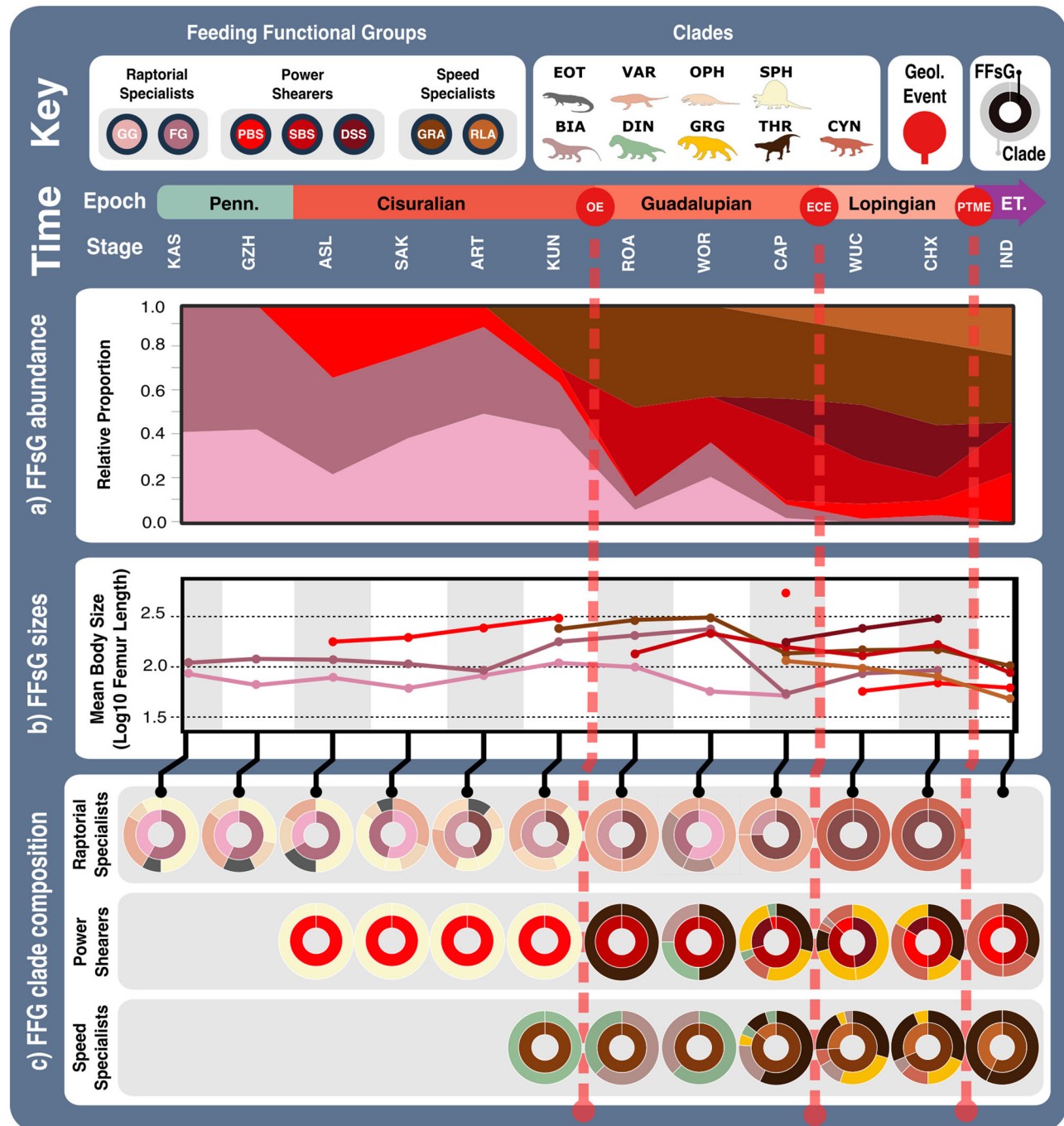

**Fig. 6 | Synapsid carnivore feeding functional subgroups through the late Palaeozoic. a** Relative abundance through time of different feeding functional (sub) groups. **b** Mean body sizes for each feeding functional subgroup through time. **c** Composition of each functional feeding group by functional subgroup and clade per time bin. Incorporates unsampled lineages using ancestral trait estimation of overall jaw shape and linear discriminant analysis for FFsG classification. Key geological events shown. Epochs are colour coded by period: Carboniferous (green), Permian (orange), and Triassic (purple). N = 122. ART Artinskian, ASL Asselian, BIA Biarmosuchia, CAP Capitanian, CHX Changhsingian, CYN Cynodontia, DIN Dinocephalia, DSS Deep shearing specialist, ECE End-Capitanian extinction, EOT Eothyrididae and assorted

Casesauria, ET Early Triassic, FFsG feeding functional subgroup, FG forceful gripper, GG gracile gripper, GRA grip and rip attacker, GRG Gorgonopsia, GZH Gzhelian, IND Induan, KAS Kasimovian, KUN Kungurian, OE Olson's extinction, OPH Ophiaco-dontidae, PBS Power bite specialist, Penn Pennsylvanian, PTME Permo-Triassic mass extinction, RLA Rapid light attacker, ROA Roadian, SAK Sakmarian, SBS Shearing bite specialist, SPH Sphenacodontia (non-therapsid), THR Therocephalia, VAR Varanopidae, WOR Wordian, WUC Wuchiapingian. Biarmosuchia, Dinocephalia and Therocephalia silhouettes by Dmitry Bogdanov (vectorized by T. Michael Keesey); all other silhouettes created by S.A.S., but some are vectorised from artwork by Felipe Alves Elias (https://www.paleozoobr.com/), available for academic use with attribution.

ecological dynamics, highlighting the growing diversity of synapsid carnivory and stark shift in feeding ecology across the early-middle Permian transition with the evolution of therapsids from basal synapsids (Fig. 1a, c).

Rising shape (Fig. 5a) and functional (Fig. 5b) disparity through the Late Carboniferous illustrates the diversification of basal synapsids, with

a notable jump in disparity driven by sphenacodontids at the onset of the Permian helping to generate most of the total morphofunctional diversity of basal synapsids (Figs. 4 and 5). Macroevolutionary modelling highlights these patterns as the 'trend' model best fits the patterns of disparity seen in basal-most synapsids and sphenacodontians (Supplementary Table 11). Full basal synapsid FFsG diversity was present by the

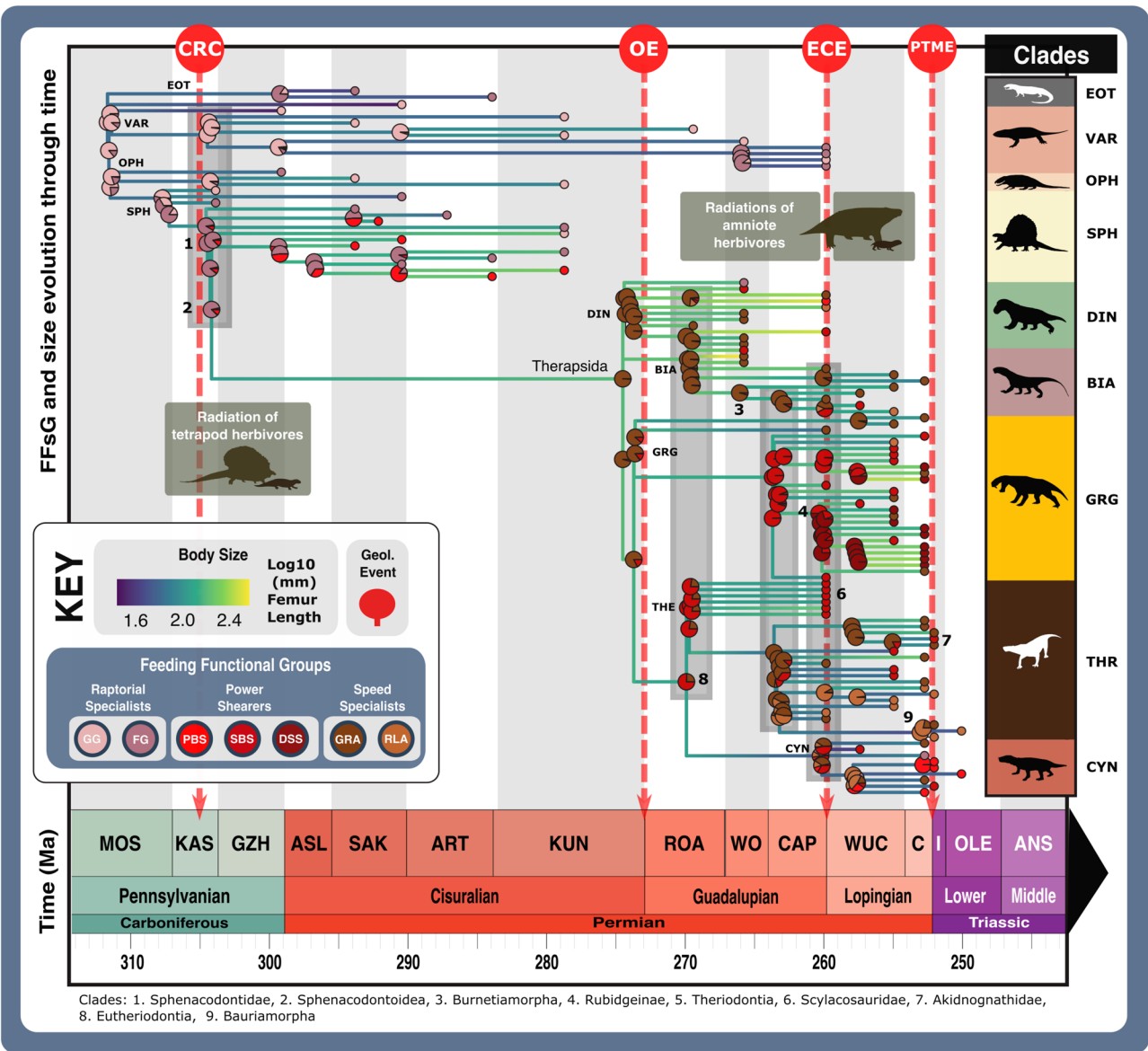

Clades: 1. Sphenacodontidae, 2. Sphenacodontoidea, 3. Burnetiamorpha, 4. Rubidgeinae, 5. Theriodontia, 6. Scylacosauridae, 7. Akidnognathidae, 8. Eutheriodontia, 9. Bauriamorpha

**Fig. 7 | Synapsid carnivore ecomorphological evolution through the late Palaeozoic.** Feeding functional subgroup states cross the carnivorous synapsid phylogeny with reconstructed ancestral character state likelihoods based on mean recovered states under equal, symmetrical, asymmetrical, and all rates different models of character transition denoted by pie charts at node positions. Positions of key clades indicated by numbers in bold across the phylogeny. Pulses of diversification highlighted with shaded boxes (grey for carnivorous synapsids and pale green for tetrapod herbivores). Body size represented by Log10(mm) femur length, with branch colour denoting low or high values (see scale). Key geological events shown. N = 122. ART Artinskian, ASL Asselian, BIA Biarmosuchia, CAP Capitanian, CHX Changhsingian, CYN Cynodontia, DIN Dinocephalia, DSS Deep shearing specialist,

ECE End-Capitanian extinction, EOT Eothyrididae, ET Early Triassic, FFsG feeding functional subgroup, FG forceful gripper, GG gracile gripper, GRA grip and rip attacker, GRG Gorgonopsia, GZH Gzhelian, I Induan, KAS Kasimovian, KUN Kungurian, MOS Moscovian, OE Olson's extinction, OPH Ophiacodontidae, PBS Power bite specialist, PTME Permo-Triassic mass extinction, RLA Rapid light attacker, ROA Roadian, SAK Sakmarian, SBS Shearing bite specialist, SPH Sphenacodontia (non-therapsid), THR Therocephalia, VAR Varanopidae, WO Wordian, WUC Wuchiapingian. Biarmosuchia, Dinocephalia and Therocephalia silhouettes by Dmitry Bogdanov (vectorized by T. Michael Keesey); all other silhouettes created by S.A.S., but some are vectorised from artwork by Felipe Alves Elias (https://www.paleozoobr.com/), available for academic use with attribution.

Asselian with the evolution of both raptorial specialist subgroups and the first power shearers (PBSs) through the Carboniferous-Permian transition (Figs. 6 and 7). Phylogenetic estimation of FFsG evolution suggests that FGs evolved from GG raptorial specialists during the Kasimovian, identifying this stage as the beginning of synapsid specialisation as terrestrial carnivores (Fig. 7). Interestingly, this interval of carnivorous synapsid trophic diversification coincides with the Carboniferous Rainforest Collapse (CRC), which saw the decline of the widespread 'coal swamps'[71,72].

The onset of the early Permian saw FFsGs become increasingly sorted by size and clade, as basal synapsids became largely restricted to

the GG raptorial specialists, while larger sphenacodontians dominated the FG and PBS subgroups (Figs. 6 and 7 and Supplementary Fig. 7). This sorting manifests in morphospace and disparity patterns. MO was largely static from the Sakmarian onwards but saw growing separation between highly robust sphenacodontids and other basal synapsids except in the Artinskian (Fig. 4). Falling functional disparity through the Artinskian saw sphenacodontians functionally overtaken by basal-most synapsids (varanopids and ophiacodontids) in the Kungurian (Fig. 5b). Therapsids were likely present in the early Permian but remain largely unknown from this interval[42]. Phylogenetic estimation suggests these hidden therapsids were GRA speed specialists that reached comparable

sizes to coexisting large PBS sphenacodontids (Fig. 7). Though this may be an artefact stemming from the traits of Roadian therapsids, it hints at a prominent therapsid role in the Kungurian carnivore guild. Nonetheless, therapsids did not become the predominant carnivorous synapsids until after Olson's extinction at the end of the Cisuralian, and the extinction of all basal synapsids except varanopids and caseids[29] (Figs. 6c and 7). Surviving varanopids remained raptorial specialists but developed more powerful jaw capabilities, expanding their MO following Olson's extinction (Figs. 4 and 6).

The first confirmed appearance of therapsids in the Roadian (Fig. 4) saw a turnover and increase in FFG diversity as new speed specialists and power shearers emerged, replacing the assortment of raptorial specialists and PBS that dominated through the early Permian (Figs. 6 and 7 and Supplementary Fig. 8). This marked the major shift in synapsid carnivory from gripping and crushing to inflicting heavy tissue damage (Fig. 3). The earliest known therapsids possessed fairly robust jaws that featured more emergent coronoid processes, placing these taxa within the central regions of both morphospaces, adjacent to earlier sphenacodontians (Figs. 1 and 4). Biarmosuchians span the divide between basal synapsids and later therapsids, exhibiting disparate robust and gracile morphologies (Figs. 1, 4 and 6), making them the only clade to feature in all FFGs (Fig. 2). Wordian dinocephalians and biarmosuchians pulled overall synapsid MO into new territory through their increased jaw robusticity and more powerful musculature, establishing a new core for synapsid MO for the remainder of the Permian (Fig. 4). Increasing disparity in the Roadian and Wordian illustrate two pulses of diversification in basal therapsids and theriodonts, respectively (Fig. 5). Varanopid basal synapsids, basal therapsids and theriodonts were all present in the Capitanian, adding much diversity to the terrestrial carnivore guild (Fig. 4). When unsampled lineages are represented using phylogenetic trait estimation, the high ecological diversity of Capitanian carnivorous synapsids becomes apparent as all FFsGs are present during this stage (Fig. 6a).

The End-Capitanian extinction event (ECE) led to broad shifts in the taxonomic composition of the FFsGs (Figs. 4 and 5 and Supplementary Table 12). Dinocephalians and most biarmosuchians perished, but their morphospace was quickly reoccupied by surviving theriodonts in the late Permian, preserving large power shearers and speed specialists across the middle-late Permian transition (Figs. 4, 6 and 7). FFsG diversity remained high in the Lopingian as highly specialised DSS and RLA carnivores diversified, reflecting contrasting hyper- and hypo-carnivorous specialisations (Fig. 6). However, the ECE saw FFGs became increasingly sorted by clade as gorgonopsians and therocephalians became the predominant power shearers and speed specialists, respectively (Fig. 6). This divergence is captured by the increase in jaw disparity through the late Permian (Fig. 5), which marks the evolution of more robust gorgonopsians and gracile therocephalians, alongside new durophagous cynodonts (Figs. 1, 4, 7 and Supplementary Fig. 8). Basal cynodonts initially appeared within central areas of morphospace but shifted their MO through the late Permian by evolving extremely robust jaws with large coronoid processes to optimise posterior MA (Figs. 1 and 2). This resurrected the PBS subgroup (Fig. 6) but the greater focus on durophagy likely reflects more generalist, perhaps even omnivorous diets in late Permian cynodonts. These cynodonts and a handful of therocephalians preserved much of carnivorous synapsid feeding functional diversity through the Permo-Triassic Mass Extinction (PTME) (Figs. 2 and 6), with only the raptorial specialist feeding group and the DSS power shearer subgroup completely disappearing (Figs. 6 and 7 and Supplementary Fig. 8). Induan therocephalians exhibited greater feeding functional diversity than contemporaneous cynodonts, which were limited to the power shearer FFG (Fig. 6). These survivors were generally much smaller than their Changhsingian counterparts (Fig. 7 and Supplementary Table 13) showcasing the recognised 'Lilliput effect' across the PTME[73].

## Synapsid carnivore specialisation and predator-prey interactions

As secondary and tertiary consumers, carnivores form the upper echelons of the food chain. Their diversity and ecologies are therefore heavily controlled by prey diversity, size, and abundance, which constrain the ecospace available for niche partitioning and so influence levels of intraguild competition[24,35,74,75]. Consequently, it is unsurprising that pulses of synapsid carnivore diversification closely correspond with radiations of tetrapod herbivores through the Carboniferous-Permian transition, and synapsid and parareptile megaherbivores in the mid-late Permian[1,76] (Fig. 7).

Early carnivorous synapsids were small, GG raptorial specialists with lightly built jaws optimised for speed, reflecting suitability for meek but slippery prey such as comparatively smaller insects, fish, and tetrapods (Figs. 5 and 7). The likely origination of the FG raptorial specialist subgroup at the beginning of the Kasimovian represents an enhancement in jaw robusticity and anterior biting efficiency (Figs. 2 and 7 and Supplementary Fig. 6a) that enabled these new raptorial specialists to better grasp, pierce and hold prey. Bite force has been found to scale positively with prey hardness and size in extant terrestrial reptiles[77,78], indicating that these new predators were likely feeding on larger prey and subject to greater stresses on their jaws during prey capture/consumption (Fig. 3). The proliferation of FG feeding functionality among eothyridids, ophiacodontids, and sphenacodontians during the Kasimovian coincides with the first diversification of herbivorous tetrapods (Fig. 7) such as diadectids, captorhinids, and edaphosaurids[76,79,80]. These new prey possessed broad trunks and robust limbs[27], making them a fleshier and so more calorific meal for predators, creating new selective pressures for terrestrial carnivores. As terrestrial herbivores became larger across the Carboniferous–Permian transition[42], carnivorous synapsids also grew in size[81] and enhanced their jaw capabilities. The emergence of the PBS power shearer sphenacodontids in the Asselian with larger overall sizes and strong jaws (Figs. 4, 6 and 7) marks the first evolution of fully-terrestrial hypercarnivores; greater anterior and posterior MA supported more powerful killing bites and easier dismemberment of the prey during feeding, while increased robusticity, particularly at the symphysis, reinforced the jaw against more rigorous stresses associated with extended interactions during prey capture (Figs. 2–4). Positional changes in the external jaw adductor musculature of these larger sphenacodontians provided additional stabilisation as well as power during jaw action, enabling these carnivores to better resist stresses associated with grappling relatively large struggling prey[56]. The rise of robust predators specialised to feed on other large tetrapods through the Carboniferous-Permian transition heralds a jump in the dynamism of terrestrial tetrapod predatory interactions (Figs. 6 and 7).

Therapsid jaw evolution appears closely linked to increasing predator-prey antagonism. The GRA speed specialist functionality emerged in therapsids during the Kungurian, having evolved from an FG raptorial specialist last common ancestor with sphenacodontids in the Kasimovian (Fig. 7 and Supplementary Fig. 9). Despite being highly derived, *Tetraceratops insignis* is close to the therapsid stem[82] and offers a potential glimpse of how GRA therapsids evolved from FG sphenacodontoids. Strong jaw robusticity, and moderate biting efficiency and speed suggest *Tetraceratops* was an active, mid-sized predator (Fig. 1). The GRA functionality builds on these characteristics, thereby supporting more vigorous interactions with prey that suggests therapsids evolved as more active predators of other terrestrial tetrapods (Figs. 4–7). Optimisation for inflicting damage (Fig. 5) potentially reflects the heightened combativeness and diversity of their prey[83,84]. Predators generally prefer relatively smaller prey and jaw mechanics suggest that non-mammalian synapsid predators conformed with this preference[67]. Prior to the middle Permian, carnivorous synapsids enjoyed a considerable size advantage over most potential terrestrial prey but the evolution of large, robust dicynodonts and pareiasaurs in the middle Permian saw herbivores close this gap with prevailing large synapsid carnivores[42,81]. The evolution of new prey that could better resist and injure attacking predators, would have increased the imperative to subdue them quickly and so gives some indication of how therapsid power shearers evolved from GRA speed specialists.

Hypercarnivory is well represented across therapsid trait evolution; dinocephalians, biarmosuchians and gorgonopsians evolved interdigitating incisors, enlarged canines, highly developed reflected laminae and robust symphyses, alongside larger body sizes[54]. Gorgonopsians also evolved differential patterns of tooth replacement across their toothrow to maintain shearing efficacy, and propalinal jaw articulation to enable wider gapes[53,54,85]. Furthermore, increased robusticity, particularly in dinocephalians and rubidgeine gorgonopsids[52,86] (Figs. 1 and 2 and Supplementary Fig. 4), supported greater resistance to internal and external loads during prey capture and highlights their specialisation to tackle more robust prey.

### Ecometric patterns within carnivorous synapsid assemblages

Ecological similarity between multiple, closely related, sympatric lineages can produce strong competitive pressures[87]. Therefore, the close relatedness of different carnivorous synapsids in successive faunas implies that intra-guild competition was a powerful selective pressure and potentially a driving force in their diversification through the Palaeozoic. Growing functional feeding group diversity and the emergence of new speed specialist and power shearer subgroups (Figs. 6 and 7), may signal growing evolutionary pressures on terrestrial carnivores across the Permian. Assemblage-level patterns of FFsG and body size (log10 femur length) distributions enable examination of potential niche overlap through the late Palaeozoic and earliest Triassic (Fig. 8 and Supplementary Data 10). It should be noted that these patterns represent a limited snapshot of late Palaeozoic assemblages, selected for their relatively rich carnivore communities, which may not reflect trends across broader geographic and/or temporal ranges.

Despite some declines through the latest Guadalupian and PTME, there is a punctuated expansion of trophic diversity across carnivorous synapsid communities through the late Palaeozoic. The potential for trophic niche overlap appears to have grown through the Permian. Clade-based sorting by FFG and size was strong at an assemblage level, and strongest in

the early basal synapsid-dominated assemblages of the El Cobre Canyon and Arroyo[88] Formations (Fig. 8). The prevalence of taxa sharing FFsG functionalities increased through time and was generally quite high in the therapsid-dominated assemblages of the mid-late Permian. Clade-based size tiering remained fairly strong, except across the Guadalupian-Lopingian transition (Figs. 6b and 8); there was reduced size and FFG differentiation across the ECE and the emergence of theriodont-dominated carnivore assemblages. Stronger size-stratification re-emerged through the Lopingian and restored a pattern seen in earlier assemblages in which raptorial specialists were the smallest carnivores and power shearers are the largest. Russian Lopingian assemblages generally show stronger size tiering than their African counterparts perhaps reflecting regional differences in ecosystem structures or fossil sampling. The *Lystrosaurus declivis* Assemblage Zone[89] captures the miniaturisation of synapsids following the PTME[73] and supports minimal changes to FFsG and relative size distributions from the latest Permian (Figs. 6 and 8).

Size distributions show that certain clades monopolised large apex predator niches through certain intervals; sphenacodontids in the early Permian, dinocephalians through most of the middle Permian, and gorgonopsians in the late Permian (Fig. 8). Basal synapsid clades converged towards larger sizes through their span as dominant terrestrial carnivores[81], whereas therapsids diversified across both smaller and larger sizes[90], emphasising body size as a key aspect of therapsid ecological variation (Fig. 8 and Supplementary Fig. 8). Niche differentiation in carnivorous therapsids appears primarily driven by varying body size rather than jaw morphofunction (Fig. 8). Body size ranges contracted through Olson's extinction, the ECE, and the PTME in a recurrent pattern of size reductions through intervals of environmental instability in non-mammalian synapsids[4]. Size-based sorting of clades across and within FFsGs also weakened through these events (Figs. 6–8 and Supplementary Fig. 8).

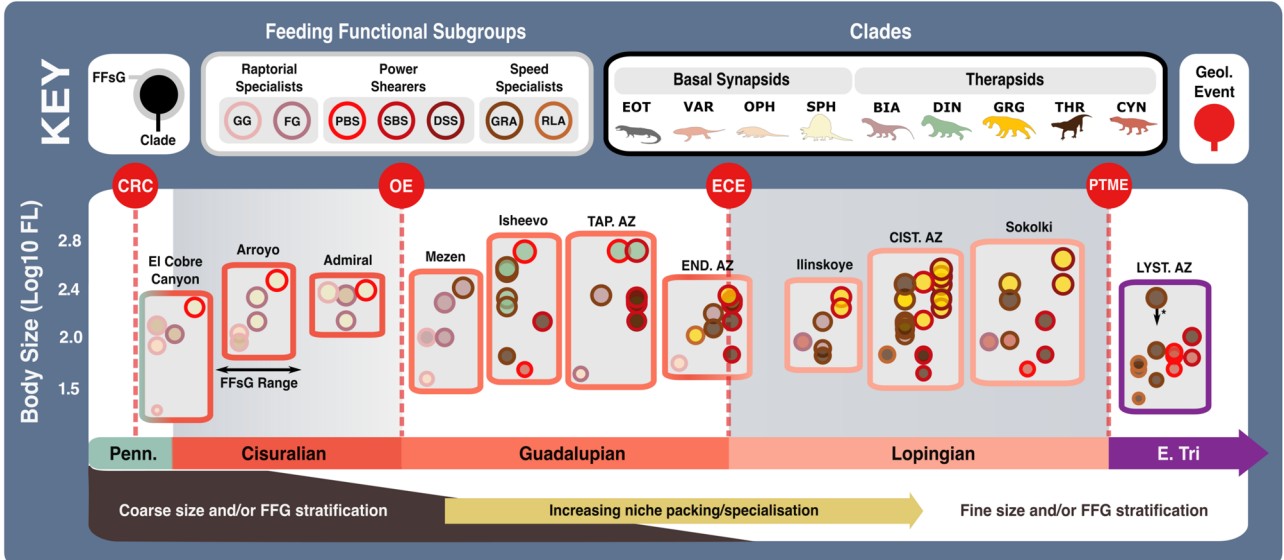

**Fig. 8 | Feeding functional subgroup and size differentiation within carnivorous synapsid assemblages through the late Palaeozoic.** The feeding functional subgroup classifications and size of carnivorous synapsids within late Palaeozoic fossil assemblages, illustrating potential ecological similarity and changes in niche differentiation. Body size represented by Log₁₀(mm) femur length. *Size based on Permian specimen as Early Triassic specimen with complete femur length measurement could not be sourced—Early Triassic specimens are typically smaller owing to Lilliput effect across the PTME[73]. Key geological events shown. Epochs are colour coded by period: Carboniferous (green), Permian (orange), and Triassic (purple). N = 81. BIA Biarmosuchia, CIST. AZ *Cistecephalus* Assemblage Zone[149], CYN Cynodontia, DIN Dinocephalia, DSS Deep shearing specialist, ECE End-Capitanian extinction, END. AZ *Endothiodon* Assemblage Zone (*Lycosuchus*—

*Eunotosaurus* subzone)[148], EOT Eothyrididae, E. Tri Early Triassic, FFsG Feeding functional subgroup, FG Forceful gripper, FL Femur length, GG Gracile gripper, GRA Grip and rip attacker, GRG Gorgonopsia, Lcm Locomotion, LYST. AZ *Lystrosaurus declivis* Assemblage Zone[89], OE Olson's extinction, OPH Ophiaco-dontidae, PBS Power bite specialist, Penn Pennsylvanian, PTME Permo-Triassic mass extinction, RLA Rapid light attacker, SBS Shearing bite specialist, SPH Sphenacodontia (non-therapsid), TAP. AZ *Tapinocephalus* Assemblage Zone (*Diictodon*—*Styracocephalus* subzone)[147], THR Therocephalia, VAR Varanopidae. Biarmo-suchia, Dinocephalia and Therocephalia silhouettes by Dmitry Bogdanov (vectorized by T. Michael Keesey); all other silhouettes created by S.A.S., but some are vectorised from artwork by Felipe Alves Elias (https://www.paleozoobr.com/), available for academic use with attribution.

The present categorical approach likely underestimates ecomorphological differences as the FFGs mask subtle differences in jaw functional morphology. However, the increasing occurrence of sympatric taxa with shared FFsG functionality and similar body sizes tentatively supports high ecomorphological similarity and so strong potential niche overlap and intra-guild conflict among carnivorous therapsids (Fig. 8).

## Potential niche partitioning and intraguild dynamics among carnivorous synapsids

Sympatric taxa within the same ecological guild are generally expected to engage in niche partitioning, which enables them to minimise competitive pressures and so maximise their feeding efficiency through morphological and/or size differentiation[24,91]. Yet, coexisting carnivorous synapsids exhibited strong ecomorphological similarity for much of the Permian (Fig. 8). Consequently, their diversification through most of the mid-late Permian appears driven by niche packing rather than niche expansion[92]. The origination of new FF(s)Gs serve as clear examples of niche expansion[92] that generally coincide with significant extrinsic changes such as the radiation of new prey and climatic events[93] (Figs. 6 and 7). Nonetheless, reduced size differences between sympatric taxa in successive assemblages suggest subdivision of existing niche space through body size (Fig. 8). Niche packing is linked to high net primary productivity, which supports the abundant resources required to enable the stable coexistence of many ecologically similar taxa and/or more complex ecosystems that permit the increased subdivision of existing niche space[94,95]. Therefore, Lopingian ecosystems may have been highly productive and heterogenous[96], given that they hosted such rich communities of ecologically conservative carnivorous therapsids.

Though limited, we find some examples of ecomorphological divergence suggestive of niche partitioning and potentially ecological displacement[19]. Ophiacodonts and sphenacodontians diverged from shared forceful gripper jaw functionality as sphenacodontians grew larger and developed more robust jaws through the Carboniferous-Permian transition (Figs. 6–8 and Supplementary Fig. 8). Therocephalians and gorgonopsians similarly diverged through the Guadalupian-Lopingian transition[97] from sharing the hypercarnivorous SBS eco-functionality as highlighted by their modified jaw adductor musculature and articulation to support powerful bites and resist disarticulation during prey capture[54,63,98]. Lopingian gorgonopsians predominantly diversified across hypercarnivorous power shearer niches, although some smaller, more gracile taxa such as *Aelurosaurus felinus* and *Lycaenops ornatus* demonstrate that gorgonopsians maintained a presence within the the speed specialist FFG (Supplementary Data 3 and 7). Therocephalians expanded mostly across the speed specialist FFG and into more mesocarnivorous niches (Figs. 4, 6–8 and Supplementary Fig. 8b). Therocephalians and gorgonopsians became further stratified by size as they were segregated between FFGs (Figs. 4, 6–8 and Supplementary Fig. 8). Size differences between these clades[97] are more pronounced in cases where sympatric taxa belong to the same FFG, further exemplifying size as a key aspect of therapsid niche differentiation (Figs. 6 and 8). Theriodont emergence in the middle Permian also precipitated apparent niche adjustments in dinocephalians and biarmosuchians that may have contributed to their respective extinctions. Both clades vacated the newly contested ecospace as dinocephalians became larger, while biarmosuchians became smaller and exclusively speed specialists (Figs. 4, 6–8 and Supplementary Fig. 8). Becoming larger would likely have heightened dinocephalian sensitivity to eco-environmental changes and so supported their extinction in the ECE. Whereas biarmosuchian specialisation may have limited their adaptability when therocephalians and gorgonopsians radiated into similar, small speed specialist niches in the latest Capitanian, encouraging the decline in biarmosuchian disparity and FFG prominence (Figs. 4, 6c–8).

Feeding morphology and body size are consistent avenues of variation for niche partitioning through the Phanerozoic[31,99–101], but it seems not especially among carnivorous therapsids through much of the mid-late Permian (Fig. 8). Body size appears more significant for ecological differentiation than jaw morphofunction, which is somewhat understandable given the limited mechanical variation of most prey tissues/materials. However, assemblage-level patterns of size variation among therapsids hint at further alternative means of niche divergence. Minimal ecomorphological differences, particularly in size, between coexisting therapsid carnivores echo patterns among extant mammalian carnivores in rich communities as seen in the African Serengeti[102] rather than other rich non-mammalian predator communities from deep time[103]. These minimal differences have been linked to limited prey escape behaviour and spatial distributions driving particular predator attack behaviour[102]. The arid, savannah-woodland mosaic environment of the modern Serengeti hosts a high diversity and abundance of prey, further highlighting the potential richness of some Lopingian ecosystems[96,102]. However, the rich mammalian carnivore communities found in the Serengeti and elsewhere also demonstrate complex patterns of antagonistic interference competition[3,24,25]. Larger mammalian carnivores tend to enjoy less hunting success than smaller competitors but are better kleptoparasites, being more successful in interspecific battles over carcasses[3,102,104]. This drives smaller sympatric predators to pursue ecological differentiation by adjusting their spatiotemporal foraging preferences[3,25,102,104,105]. Therefore, heightened jaw morphofunctional similarities offset by increased size differentiation may imply growing behavioural complexity among therapsids through the Permian. Scleral ring and orbit size variation indicate diverse diel activity patterns across non-mammalian synapsids[106] supporting their ability to temporally partition carnivorous niches. Moreover, ecological differentiation could have been primarily realised through postcranial aspects of skeletal anatomy.

## Impacts of synapsid postcranial evolution on trophic ecology

Carnivore foraging and prey preferences are closely linked to their speed, agility, and endurance[104,107]. Whether a predator employs ambush or pursuit hunting techniques across a small or large range of territory is linked to their locomotor capabilities. The forelimbs may also be involved in prey capture, being used to pin down and stabilise prey, thereby easing the stresses and strains on the jaws[55,65]. Furthermore, by shaping these behaviours, locomotor traits affect how sympatric carnivores vary their habitat and foraging preferences to differentiate their niches and minimise intra-guild conflict[24,25,104,105,107].

Axial and appendicular anatomy offer some indication of locomotor capabilities and were key loci of evolutionary change for synapsids through the late Palaeozoic. Synapsid axial and appendicular evolution is linked to advances in locomotor efficiency[61] and denotes their growing propensity for terrestrial lifestyles. Trends towards reduced lateral undulation during ambulation, increasingly erect postures, forward-facing feet, and increasingly parasagittal gaits point to enhanced stability and efficiency during locomotion[50,61,108–112]. Remodelling of the pectoral girdle and shoulder joint permitted a wider range of forelimb motion supporting manoeuvrability across rough terrain and wider utility beyond locomotion, despite the forelimbs retaining a sprawling position[113]. Greater morphological disparity across the basal synapsid-therapsid transition further demonstrates the increasing utility of the forelimb in Palaeozoic synapsids[114,115]. Growing locomotor capabilities likely enabled new foraging and prey capture techniques by synapsid carnivores.

Shifting emphasis from catching to killing prey across the basal synapsid-therapsid transition (Figs. 3 and 7) may reflect a shift from passive 'sit and wait' hunting towards an 'active search' approach characterised by seeking out prey[116]. Active search hunting is linked to greater locomotory efficiency[117] and typically employed by larger predators that track prey over distance[102,116,118,119]. Larger sizes (Figs. 6–8) combined with further auditory[120–122] and olfactory modifications[123–125] may have supported such long-distance tracking capabilities among some therapsid carnivores. Increased energetic constraints on larger carnivores[126,127] would have driven more antagonistic interference competition, echoing size-based intra-guild relationships in present faunas[24,104], creating strong selective pressure for further ecological differentiation via improved locomotory and foraging efficiency[102]. Postcranial morphology[108,109,128] and fossil trackways[129] point to greater ambulatory speeds among carnivorous (theriodont) therapsids and

suggest they were capable of small bursts of acceleration, supporting the possibility of felid-like ambush hunting in some carnivorous therapsids[102]. Therapsid torso morphology pulled their centre of mass anteriorly relative to basal synapsids, towards the thoracic region and the heavily built, splayed forelimbs, helping to anchor therapsid predators when attacking prey with their jaws[116]. This stabilisation further supported lateral head movements, facilitating slashing attacks against prey[130], and emphasising the offensive utility of the jaws (Fig. 3). Additional emphasis on damaging and so quickly incapacitating prey may also partially reflect the growing mobility of prey when considered alongside tetrapod locomotor evolution through the late Palaeozoic[112], as quick kills would have limited opportunities for prey to escape as well as cause injury to the attacking predator.

Patterns of trophic differentiation observed here may relate to differences in locomotory ability as synapsids evolved a growing aptitude for terrestrial lifestyles[15]. Differences in postcranial anatomy may indicate that FFG divergence between ophiacodonts and sphenacodontians reflects divergent preferences for semi-aquatic and terrestrial habitats, respectively[27,131] (Figs. 4 and 6 and Supplementary Fig. 8). Differences in locomotor capability may have also shaped aforementioned examples of potential therapsid niche partitioning. Theriodonts possessed greater locomotory and appendicular capabilities than biarmosuchians and dinocephalians[61,102], which may have supported their survival through the ECE and subsequently allowed them to overtake surviving biarmosuchians (Fig. 6). Advances in locomotor ability are also associated with respiratory and metabolic changes linked to the origin of synapsid endothermy[8,114,132,133]. Endothermy allows mammals to regulate and maintain high body temperatures independently of external environmental conditions, unlike most reptiles, which are ectothermic. Dating the emergence of synapsid endothermy remains challenging, but it could be that the differentiation of gorgonopsians, therocephalians and cynodonts in the Lopingian (Figs. 6–8) may partially stem from early endothermy in eutheriodonts[133,134]; the resulting adaptive flexibility may have enabled eutheriodonts to diversify across smaller sizes and mesocarnivorous niches, as well as survive the PTME. However, much study is needed to clarify the presence of endothermy across Permian therapsids.

Despite growing similarity to mammals, non-mammalian synapsids still lacked the axial and appendicular flexibility[50,61,108,111,112,114] and likely metabolic capabilities[132–134] to pursue or grapple prey exactly as seen in modern mammalian carnivores[65,66,102], so drawing exact parallels is problematic. Further detailed study is required to better assess non-mammalian synapsid locomotory functional diversity and properly evaluate the ecological context.

### Carnivorous synapsid macroevolution and the complexification of terrestrial ecosystems

Heightened overall ecomorphological diversity (Figs. 4–7), yet reduced similarities at an assemblage-level (Fig. 8) among synapsid carnivores through the late Palaeozoic appear to reflect the growing complexity of terrestrial trophic dynamics as tetrapods became more proficient on land[1,11,14,15]. The onset of climatic trends towards greater aridity and seasonality in the Kasimovian saw increasing environmental heterogeneity through the CRC[71,72]. Early amniotes were closely associated with dry, upland environments[28,135] and the spread of drier habitats through the CRC favoured amniotes, likely encouraging their diversification[5,9,42]. The first appearance of tetrapod herbivores during this time made more energy readily available to terrestrial ecosystems by tapping into the productivity of land-based vegetation, pulling tetrapod food chains further inland and spurring terrestrial trophic network complexification[7,76]. Prevalent size stratification between larger PBSs and smaller FGs across the Carboniferous-Permian transition illustrates greater niche diversity in the terrestrial carnivore guild and marks the addition of new tiers to terrestrial trophic networks (Fig. 6 and Supplementary Figs. 7 and 8).

The growing size and locomotory proficiency of terrestrial tetrapods extended their geographic ranges, allowing them to exploit a wider array of terrestrial resources; herbivores could browse on vegetation over wider areas and carnivores could forage over further distances[119]. Consequently, terrestrial trophic networks saw an increase in the energy flowing through them, permitting increasing sizes in terrestrial carnivores and herbivores[42], as well as greater subdivision of resources, which is captured in the rising FFsG diversity and community richness of carnivorous synapsids through the mid-late Permian (Figs. 6–8). Further evidence of greater energy in terrestrial trophic networks comes from the shift in carnivorous synapsid jaw optimisation from grasping and holding prey to injuring and subduing them (Figs. 3 and 7). Larger and more mobile prey could better flee or fight back, and thus the apparent drive to incapacitate prey quickly signals the heightened volatility of predator-prey confrontations. Further mesocarnivorous specialisations in late Permian eutheriodonts highlight a crowded terrestrial carnivore guild and illustrate further complexification of terrestrial trophic networks.

Here we quantitatively show that the predominant terrestrial carnivores of the late Palaeozoic underwent a dramatic shift in their jaw morphofunctionality across the early-middle Permian transition, which when considered alongside patterns of size evolution, reflects a broad shift in the trophic interactions of terrestrial tetrapods. The evolution of sphenacodontian hypercarnivores in the Carboniferous and neotherapsids in the Capitanian mark the growing antagonism of tetrapod predator-prey interactions and increasing dynamism within terrestrial trophic networks, as climatic and wider morphofunctional shifts drove revolutions in terrestrial tetrapod life on land. As synapsid carnivores adapted to life on land in more heterogeneous habitats offering a growing diversity of resources, they were subject to an array of selection pressures. The ensuing patterns of niche diversification among middle and late Permian synapsid carnivores illustrate the first emergence of complex terrestrial ecosystems and provide remarkable parallels with living large mammalian carnivores.

## Methods

### Taxonomic sampling and data collection

A list of all valid synapsid carnivore taxa and their stratigraphic ranges, across the Late Carboniferous to Early Triassic, was assembled using the published dataset of Benton et al.[136,137], and more recent literature to incorporate subsequently described taxa and taxonomic and stratigraphic revisions (Supplementary Data 4) following the approach of Singh[138]. Absolute age assignments were to stage level and based on the 2023 version of the International Chronostratigraphic Chart[139]. This list was used to compile a collection of complete lower jaw images from photographs and specimen drawings from the literature alongside photographs taken during museum collection visits[138] (Supplementary Data 11). This study was conducted at genus level to maintain a balance between the availability of data and confidence in taxon diagnosis, as most genera are monospecific[138]. Consequently, a single specimen was generally used per genus, precluding assessment of intraspecific variation, which would require significantly more sampling. However, multiple specimens per genus were used where multiple species were available; basal synapsids typically possess much greater species diversity per genus than their therapsid relatives, and while this may bias our morphometric analyses, we would otherwise ignore much synapsid diversity. The genera with multiple species are: *Dimetrodon (D. grandis, D. limbatus, D. loomisi, D. milleri,* and *D. natalis), Haptodus (H. garnettensis* and *H. baylei), Ophiacodon (O. uniformis, O. mirus,* and *O. retroversus), Sphenacodon (S. ferocior* and *S. ferox), Alopasaurus (A. gracilis* and *A. tenuis), Inostrancevia (I. alexandri* and *I. latifrons),* and *Sauroctonus (S. parringtoni* and *S. progressus)*[138].

Maximum femur length (Supplementary Data 3) was used as a measure of overall body size, as this measurement is widely available from published literature, enabling more comprehensive study of size dynamics[138,140] (Supplementary Data 12). Data were augmented using a multi-rate Brownian motion model of phylogenetic character reconstruction to impute femur length values for taxa without such femoral data[141,142] with the mvMORPH package[143].

This study includes a total of 122 taxa representing 110 genera. Our sample includes two eothyridids, 11 varanopids, five ophiacodonts,

15 sphenacodontians, 14 biarmosuchians, 10 dinocephalians, 27 gorgonopsians, 28 therocephalians, and nine cynodonts[138]. Omnivorous diets have been suggested for basal dinocephalians compelling their inclusion here alongside evidently carnivorous anteosaurids[144]. Damaged, distorted, and juvenile fossil material were excluded where possible. Functional data were also included from 23 extant taxa (10 reptiles, eight canids, and five felids) to help interpret non-mammalian synapsid trophic ecology.

Assemblage data was used to understand more local-scale patterns of size and FFsG differentiation (Fig. 7) and was compiled from published literature with assignments made according to the most recent studies[11,89,136,145–149]. Assemblages featuring relatively rich carnivorous synapsid diversity across the late Palaeozoic were included for comparison in this study as it was felt these sites represented more complete communities, where likely intraguild dynamics of coexisting carnivores could be better assessed. Taxa in these assemblages that were not included in the present study are sometimes represented by closely related taxa that are featured here, where potential substitute taxa were deemed similar according to published literature (Supplementary Data 9).

## Phylogeny
We generated an informal synapsid supertree following Singh[138] using a scaffold based on Mann and Paterson[150], which expanded and modified the character matrices used by Benson[151], Reisz and Fröbisch[152], and Brocklehurst et al.[153]. Additional scaffolds were used to construct the topologies of major synapsid subclades (see Supplementary Methods)[138]. Recent study has suggested that varanopids may be diapsids rather than basal synapsids, as part of a broader reorganisation of phylogenetic relationships at the amniote base[154]. However, Ford and Benson[154] stress the extreme uncertainty of this topology, and more recent anatomical study of varanopids supports their traditional inclusion as synapsids[45]. Therefore, we follow the established synapsid phylogeny. Additional taxa were added using Mesquite 3.51[155]. The topology was time-scaled using the minimum branch length (MBL) method with a minimum branch length of 0.1 myr[156] as implemented in the timePaleoPhy function of the R package paleotree[157].

## Morphometric analyses
Geometric morphometric methods use user-defined landmarks and Cartesian coordinates to capture shape variation, whereas functional morphometric methods use standardised functional measurements (SFM) (see Supplementary Methods) that reflect clear, ecologically relevant aspects of jaw function. Synapsid evolution encompasses significant changes in jaw anatomy, particularly through the transition from basal synapsids to therapsids, and the evolution of mammals[29,61,138]. Difficulties identifying homologous points of bone articulation and the poor preservation of many specimens led us to adopt a Type II landmarking regime that focuses on overall shape. A Type II approach lacks the capacity to clearly assess modular changes in jaw anatomy and ensuing patterns of mechanical evolution but provides a flexible framework to assess broader patterns of trophic ecology across nonmammalian synapsids. Our regime uses four fixed homologous landmarks, connected by four semi-landmarked curves comprised of a total of 55 semi-landmarks, placed equidistantly along each curve (Supplementary Fig. 1)[138]. Images were digitally landmarked using tpsDig2[158] and processed in tpsUtil[159] to define the semi-landmark curves. A Procrustes transformation was applied in tpsRelW[160], using the chord–min d² sliding method that restricts semi-landmark movement along a chord between its two adjacent landmarks[138]. The Procrustes transformation standardises the differences in size and orientation in the landmark data to generate aligned coordinate data. The functional morphometrics used eight functional characters (Supplementary Data 2) based on linear measurements of the jaw (Supplementary Fig. 2) taken using ImageJ[161]. These measurements capture aspects of jaw functionality from the overall jaw shape, such as the lever-arm mechanics of the jaw

musculature, articulation, and robusticity, which have been used to characterise overall jaw function and interpret feeding ecology[32,33,162] (see Supplementary Methods).

Separate principal component analyses (PCAs) were applied to the shape-aligned coordinate data and SFM to identify the major axes of form and functional variation, using geomorph[163] for the geometric data, and FactoMineR[164] for the functional character data[138] (Supplementary Note 1). The functional measurement data was centred and standardised using a z-transformation prior to the PCA to mitigate heteroscedasticity[31,138,162]. The resulting first two PC axes were used to plot morphospace occupation as they reflect the greatest aspects of variation[138]. The PC scores were combined with functional character data to generate contour plots of different aspects of jaw function across morphospace using linear interpolations via the akima package[165].

## Statistics and reproducibility
Overall differences in jaw shape and function between clades, time-bins, and FFGs were assessed via one-way non-parametric analysis of variance (NPMANOVA) using a Euclidean similarity index using all relevant PC scores and a Bonferroni correction. Significant size differences across timebins were identified using a pairwise Mann–Whitney $U$ test. Both sets of tests were carried out in PAST (version 3.24)[166] and used Bonferroni corrections to minimise Type I errors stemming from multiple comparisons.

## Calculations of disparity through time
Morphological diversity (disparity) was measured by calculating within-bin sum of variance (SOV) to assess patterns of shape and functional disparity through time. SOV is quite resistant to sampling biases and so provides robust temporal patterns of disparity[138,167]. We used a phylogenetic timeslice approach[70] applied using the dispRity[168] R package to generate SOV through time for different clades, while also incorporating unsampled lineages. All PC axes were used in the calculations with 1000 cycles of bootstrapping to provide 95% confidence intervals and rarefaction to minimum time bin sample size to account for differences in sampling per subset[138]. Shape and functional SOV were plotted to substage level alongside time-slices of each morphospace using the calibrate[169] and strap[170] R packages. Our time-bins were generated by equally dividing each stage into an upper and lower substage.

## Consensus cluster methods
To classify carnivorous synapsids into functional feeding groups based directly on their ecomorphology, we use the consensus clustering approach of Singh et al.[31] and Singh[138] and see these sources for further methodological detail. This approach requires minimal prior input or supervision and uses different hierarchical and partition clustering algorithms[171] to produce robust, objective functional feeding groups (FFGs)[31,138]. The consensus cluster approach was applied to the functional data owing to their distinct eco-functional utility which enable clearer interpretations of likely feeding behaviour relative to shape data, which carries greater phylogenetic signal and neglects important features such as such muscle attachment positions[31,138]. The SFM were used to generate a Euclidean distance matrix that was subjected to hierarchical, K-means and partitioning around mediods (PAM) clustering analyses using the 'eclust' function of the FactoExtra[172] package[138]. We used a defined cluster (K) range (2–10)[173] using gap statistic values generated from 2000 bootstrap cycles following Singh et al.[31] and Singh[138]. Results were evaluated using the 'cluster.stats' function from the fpc[174] R package using silhouette metrics to illustrate clustering performance and phylogenetic signal using external validation metrics[171] (Supplementary Tables 9 and 10). Resulting clusters were compared to generate composite groups based on classification consensus, creating the FFGs. Majority rule was used to designate the typical FFGs of clades based on the classification their taxa classified here[31,138].

The FFsG classifications were validated using a jack-knifed, linear discriminant analysis (LDA)[41]. The LDA was implemented with a jackknifing test in PAST (version 3.24)[166] using all functional PC scores, and

correctly classified 86% of taxa[138]. Classification uncertainty is typically present in taxa positioned at the margins of their respective FFsG. (Supplementary Fig. 6), indicating that these taxa likely exhibited trophic ecologies that mixed elements of the core FFsGs. This potentially highlights the troughs in the adaptive landscape of Palaeozoic synapsid carnivores and the reality that realised niches exist within a spectrum, varying considerably depending on a range of factors such as the conspecifics present and available habitat resources[138,175]. An LDA was also used to classify the FFsG of ancestral taxa based on ancestral state reconstructions of functional PC scores to plot trends in FFsG across clades and body size through time (Fig. 6)[138].

## Phylogenetic methods

The DispRity[168] R package was used to apply macroevolutionary models and test whether disparity trends followed a Brownian Motion, Early Burst, Ornstein-Uhlenbeck/Constraint, Trend, or Stasis model of macroevolution. Model fit and support was assessed using weighted Akaike Information Criterion (AIC) and log-likelihood values (Supplementary Table 11). Phylogenetic estimation of FFsG, functional PC scores, and body size for nodes across the synapsid phylogeny were carried out to better track the ecomorphological diversification of synapsid carnivores. FFsGs were reconstructed as discrete character states using maximum likelihood estimation[176,177] via the 'asr_mk_model' function of the castor[178] R package. FFsGs were estimated under models using equal rates, all rates different, symmetrical, and asymmetrical rates of character transition[179] (Supplementary Table 14), with the mean results of all models being mapped onto a time-scaled phylogeny using the strap[170] and ggtree[180] R packages (Fig. 7 and Supplementary Figs. 9–12). These mean results are used for discussion and reflect a conservative estimate of trait evolution given the general uncertainty regarding niche boundaries. Functional PC scores and body sizes (the $\log_{10}$ transformed femur length) were estimated as continuous traits using a maximum likelihood approach via the 'FastAnc' function of the phytools[181] R package. Resulting body size values were also mapped onto the phylogeny using the ggtree[180] R package (Fig. 7)[138]. The reconstructed fPC scores were used to classify the FFsG of ancestral nodes using an LDA to plot FFsG prevalence through the late Palaeozoic, and estimated body sizes were used alongside recorded taxon sizes to plot mean body size per FFsG (Fig. 6)[138].

## Reporting summary

Further information on research design is available in the Nature Portfolio Reporting Summary linked to this article.

## Data availability

The authors declare that all the data directly supporting the results of this study and underlying all figures are included within the Supplementary Data files linked to this paper. The data used in this study can be found for download at Dryad: https://doi.org/10.5061/dryad.jq2bvq8h0.

## Code availability

The authors have included R scripts to perform the analyses implemented in this study within the Supplementary Information linked to this paper and as a text file available for download at Dryad: https://doi.org/10.5061/dryad.jq2bvq8h0.

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

## Acknowledgements

Many thanks to Christine Janis, David Button, Daniela Schmidt, and Stephen Brusatte for providing constructive commentary. We thank our reviewers, David Grossnickle, Jun Liu, and Megan Whitney for their helpful comments, which greatly improved this paper. We also thank Dmitry Bogdanov, T. Michael Keesey, and Felipe Alves Elias (https://www.paleozoobr.com/) for permitting academic use of their excellent artwork. Funded in part by the NERC BETR grant NE/P013724/1 and ERC grant 788203 (Innovation) to M.J.B., NERC grant NE/L002434/1 to A.E., and NERC grants NE/P013724/1 to T.L.S., A.E., and S.A.S.

## Author contributions

S.A.S. designed the study, collected the morphological data, and conducted all the analyses. A.E. helped source jaw images and provided the femur length, stratigraphic and phylogenetic data. T.L.S. consulted on the morphometric analyses and figure composition. E.J.R. and M.J.B. provided guidance and comments on the manuscript. S.A.S. wrote the manuscript with input from all coauthors.

## Competing interests

The authors declare no competing interests.
