## [Peer Review File · Communications Biology]

Reviewers' comments:

Reviewer #1 (Remarks to the Author):

General comments to authors

Your paper is a strong contribution to our understanding of synapsid carnivore evolution. The foundation of the study is morphometric analyses of jaw shape, but you've expanded on those analyses in interesting ways, such as classifying taxa into functional feeding groups, incorporating phylogeny into some analyses, and using the results to infer predator-prey interactions. In some places I think you may be a little too assertive about your inferences of feeding behaviors and ecological interactions – inferring behaviors/ecologies from jaws (and size) alone is a bit speculative (see comments below). But that said, I find the ecology-focused aspect of this paper much more interesting and informative than similar studies that only focus on morphology or disparity patterns. And I think it'll be of interest to a broad readership.

Admittedly, my taxon expertise lies in the crown mammal portion of the synapsid tree, so I don't have any comments on specific taxa or how you classified them into feeding groups, but hopefully other reviewers/editors can evaluate that aspect of the paper. Instead, I focused my review mostly on the data and analyses. My only major concern is about how you treat the functional vs shape data, which I explain below. Most comments are meant to be constructive suggestions rather than critiques.

Feel free to contact me if you have any questions about my review.

Dave Grossnickle
dmgrossn@uw.edu

Major comments

I strongly support your use of functional traits in morphometric analyses. However, I have concerns with how you present your functional traits and analyses.

First, your 'functional' data and 'shape' data both capture very similar jaw information. The functional traits are measured using shape traits such as jaw length, jaw height, in-levers (measured to jaw margins), tooth row length, etc. (Fig S2), and these traits are all captured (along with other traits) by the geometric morphometric shape analyses (Fig S1). And this overlap is reflected in your results – the multivariate plots/disparsity patterns for shape and function look very similar (Figs. 1, 4, 5). This issue is magnified because we don't know how well your functional traits are tied to functions due to the lack of behavioral data or modern analogs for extinct synapsids (as you acknowledge, e.g. Line 79). Most of your functional traits seem to be chosen based on traits of modern taxa (based on your citations in the Supplementary Methods), but these may not be relevant to early synapsids. For instance, therian mammals chew in a fundamentally different way than earlier synapsids (e.g. Grossnickle et al. 2022 ZJLS), and thus their functional jaw traits may be very unique to those of earlier synapsids. In other words, your functional analyses may not be capturing functional information any better than your shape analyses.

This overlap between 'function' and 'shape' analyses by itself isn't necessarily problematic. But my concern is that you present them as two independent analyses (e.g. Lines 88-91) – shape vs function – which seems a bit misleading. And, the analyses feel redundant. Thus, my recommendation is to treat one analysis (maybe shape) as primary and the other analysis (maybe function) as supplemental, and describe BOTH analyses as alternative morpho-functional or ecomorphological analyses (rather than labeling them as 'shape' and 'function' analyses). This would help to simplify the

complex figures. For instance, I don't think you need stacked morphospace plots (Fig 5) for both 'shape' and 'function' (they look very similar), and you could move one to the Supplement. I think the most valuable functional results are the univariate plots in Figure 2, which I recommend keeping in the main text.

A related issue is that some of your MAMA/MPMA functional traits are based on subjective muscle insertion reconstructions (Fig S2), and I don't see an explanation on how these muscle insertion sites were reconstructed. Did you avoid reconstructing the masseter on delicate reflective laminae/angular bones of many synapsid groups (e.g. see the *Thrinaxodon* reconstruction in Lautenschlager et al. 2017 Biological Reviews)? If you reconstructed the masseter on the angular of some taxa, which ones? Did you utilize reconstructions in previous studies (e.g., Lautenschlager et al. 2017)? These types of details should be explained. I realize that paleontologists must often make subjective decisions about muscle locations (and I'm certainly guilty of it). But, at minimum, you should better explain your decisions, and you should acknowledge and address uncertainties.

The figures are attractive and packed with information. However, they rely heavily on multiple color schemes, which can be confusing. (And they might be hard for colorblind readers to interpret – have you checked that the color choices are appropriate?) For instance, Figure 6 includes color schemes for FFGs (mostly shades of red), geologic epochs (mostly shades of red), and clades, and geologic events are marked by a red circle/line. Where possible, I recommend reducing the number of colors (or color schemes) used. The colors for the geologic ages are probably least important, so maybe you could remove those colors in all the figures? You could replace them with alternating gray and white, like you use in the background of Fig 7.

You use a lot of '-ism' and '-ity' words. Maybe it's just personal opinion, but I find that style of writing harder to read. For instance, you say "using the well-established ecological utility of mandibular functionality and body size" (Line 78), which didn't make any sense to me the first time I read it. I'd probably re-phrase that to something like "using jaw functional morphology and body size, which are well-established correlates of ecological traits." Another example is "growing antagonism and dynamism ... of interactions" and "complexification" (Lines 18-20). I'd instead prefer phrases such as "an increase in antagonistic interactions" and "increased complexity". But this is just my opinion, so feel free to ignore my comment if you'd like.

You use a LOT of acronyms, which are hard for a reader to keep straight and often seems unnecessary. For instance, OE is defined and then only used once (I think) in the main text (although it's also used in the figures). And abbreviating phrases like 'morphospace occupation' as MO seems unnecessary. Having too many acronyms results in phrases like this: "... to optimise MPMA (Figs. 1–2), resurrecting the PBS FFsG" (Line 277), which won't make sense to a reader who hasn't memorized the acronyms. I recommend removing any acronyms that aren't used regularly throughout the paper.

Much of your later discussion is on ecological inferences, which I like. But an issue is that some topics are challenging to address using a limited sample of taxa that are temporally scattered over tens of millions of years. For instance, you discuss niche partitioning, which can result from the coexistence of competing taxa (Line 341). Maybe I missed it, but I don't see any discussion on evidence that species in your sample coexisted/interacted. How much temporal and geographical overlap was there among taxa in your sample? Are many of the taxa found at the same fossil localities?

I think it'd be safer to frame your discussion more broadly on 'ecological/functional diversification' rather than increased 'niche partitioning'. Your results demonstrate ecological diversification, whereas niche partitioning is just one potential cause for ecological diversification (along with increased geographical isolation, stochastic change/genetic drift, etc). If you maintain the niche partitioning discussion, then I recommend tempering your comments and maybe presenting your conclusions as

hypotheses for future testing.

Minor/line comments

Abstract: I don't think you mention jaws in the Abstract. I recommend mentioning jaw shape because 1) it's the foundation of the study and 2) otherwise lines like "specialised for either power or speed" could be confused with non-jaw related traits (e.g. locomotor speed).

Lines 9-10: The first couple of sentences could be revised for clarity. You might revise the first sentence to start with something like "Terrestrial ecosystems increased in complexity in the ...". And in the second sentence I'd expand "among these were" to "among these predators were"; otherwise "these" seems to be referring to "terrestrial ecosystems," the subject of the first sentence. Also, "than ever" implies that ecosystem complexity was greater in the Permian than at any other time in history, including more recent ecosystems. Is that what you mean?

Line 45: "they sought" implies that the tetrapods consciously tried to evolve certain ways. Their evolution was driven by natural selection, not the lineages themselves. I'd revise the sentence.

Lines 75-85: This is a nice summary/introductory paragraph. But just like in the abstract, you never really introduce that your data is based on jaw shape – you instead say "mandibular functionality" and "jaw functionality," which is confusing to the reader. Also, you could provide more evidence in this paragraph as to why/how jaw shape is correlated with diet/ecology (maybe from a biomechanical perspective) and thus makes an ideal study system.

Line 88: "mandibular". I assume based on your jaw images (Figs S1 and S2) that you measured the entire lower jaw (including post-dentary bones), correct? I recommend explicitly stating this in the text. Also, you might consider stating that you use the terms "mandible" or "mandibular" broadly to refer to both the dentary and postdentary bones – some researchers might just associate the mandible with just the dentary and/or only use "mandible" to refer to crown mammal jaws. Alternatively, you could replace "mandible" with "jaw" throughout the paper.

Line 108-110: The similarities in the morphospaces that you mention here highlights my point about the analyses being very similar and a bit redundant.

Lines 113-114 (and Fig S3): PC1 and 2 scores are already presented in Figure 1, so I think it'd be more informative to plot PC3 and 4 scores in Figure S3 (rather than 1v3 and 2v3). And rather than calling them "secondary morphospaces," I'd refer to them more specifically as something like "PC3-PC4 morphospaces" (or "PC1/2-PC3 morphospaces" if you don't follow my advice).

Lines 147-194: I like these FFGs, and I like that they're supported quantitatively by the cluster analysis and LDA. But I was underwhelmed by the references cited and additional empirical evidence to help support and validate these classifications. Maybe there simply hasn't been much work done on the diets of early carnivores (at least relative to herbivores)? You might double check the literature, including Crompton papers, and DeMar & Barghusen 1971 – I think they tested a hypothesis on the feeding behaviors associated with evolution of the coronoid process. And/or maybe you could expand your discussion on how tooth/skull morphology helps to support the inferred feeding behaviors?

Line 198: I recommend explaining "kinetic inertial biting system".

Lines 471-475: This text feels a little out-of-place in the Methods. You could move it to the

Introduction (maybe the last paragraph, Lines 75-85) or Results.

Line 494: Was "areas of muscle attachment" used as a functional trait? I don't think so, unless I misunderstand one of your functional traits? You use muscle inlevers for the MAMA/MPMA traits, but those aren't necessarily capturing areas of muscle attachment. For instance, a small muscle with an anteriorly positioned insertion site would have a small attachment area and a long inlever.

Line 503: What is the "D" in "SFMD"? I don't remember that being defined. And SFM is another example of an acronym that seems unnecessary.

Line 523: "To identify the niches of synapsid carnivores ..." seems vague and potentially misleading. I'd instead say "To classify synapsid carnivores into functional feeding groups ...".

Line 530: Can you use the geometric morphometric landmark data (or Procrustes or PC scores) as input in the cluster analysis? As I explain above, I'm skeptical that the functional data capture function any better than the shape data, so you might try analyzing the shape data and see if results are congruent with the functional data.

Line 542: You might also list the LDA's classifications, maybe in your Supplementary Dataset file.

Figure 1b: I only see these plots mentioned once in the text (Line 97), and the results can't be easily interpreted by readers, so it's unclear to me how they're important for the paper. You might either move them to the Supplement, or, if you think they're important, then add more discussion on them.

Supplementary Materials Lines 139-155: This is a nice summary, but it seems to be missing some details. I assume "MA" is mechanical advantage and is measured as the inlever divided by outlever, which is implied later, but you should explicitly state that. The MAMA and MPMA are means of multiple MA ratios, correct? I don't think you explain that here and don't mention what ratios are averaged (at least that I see), although I think I figured it out from the text in Fig S2. You should provide a little more information about which inlevers correspond to which muscles, and how you determined the inlever positions (see my comments above). In Fig S2 it looks like they're measured to the midpoints of the muscle attachment areas (but to the jaw margins and not to the actual center of the muscle areas), which is fine, but I personally would have probably measured them as perpendicular to the inferred muscle force vector. And as I mention above, the masseter position could be especially problematic because of the postdentary bones in the same region.

Figure S2: For the X/Y/Z lines, there are some abbreviations (for muscles?) that should be defined (unless I missed it): MPT, MAME, MAMP.

Supplementary dataset: I appreciate the thorough, well-organized dataset! I only see it cited on Line 498 and in the Data Availability section. It contains valuable information that readers may want to check out, so I recommend citing it where relevant, such as after mentioning femur length data (Line 445) and functional trait data (Line 493).

Reviewer #2 (Remarks to the Author):

Based on the morphometric and macroevolutionary analysis on the lower jaws of carnivorous synapsids, this paper give a new point on the ecological evolution of terrestrial tetrapod ecosystems. As far as I know, it collected the abundant data (although there are errors), I am not quite familiar with some methods applied in this paper, and I do not repeat the analysis, but as far as I know, the

major conclusion is convince. It is worth to be published after revision of the following places.

For Permian, early , middle or late are informal and should be lower case; also for lower and upper.

L110 Basal synapsids and therapsids are distinguished principally by reductions in RTL and increasing prominence of the postdentary bones in therapsids.

The postdentary bones are not prominent in some therapsids than 'pelycosaurs'. The most prominent feature is the reflected lamina. This also different what you stated later (Basal synapsids developed increasing robusticity and enlargement of the mandibular body from varanopids and ophiacodonts to (non-therapsid) sphenacodontians).

L252 The earliest known therapsids in the Roadian possessed fairly robust mandibles that lacked prominent coronoid processes, placing these taxa within the central regions of both morphospaces, adjacent to earlier sphenacodontians.

The only Rodian therapsid in your list is Phthinosaurus, which has a prominent coronoid process. Please check its morphospace!

L431 For the age, you should use the age on the latest version (2022), and change your matrix and figures.

Reviewer #3 (Remarks to the Author):

Singh et al. employ geometric morphometrics and phylogenetic comparative methods to study the evolution and occupation of different feeding niches in carnivorous basal and therapsid synapsids. They find evidence for discrimination between basal synapsid and therapsid mandibular form and function by examining a suite of functional measurements. They find increased niche partitioning within carnivorous therapsids and suggest that this disparity in function allowed for increased predator-prey interactions and may have played a role in mass extinction selection barriers and synapsid evolution more broadly. This a well-written paper that is thorough and thoughtful. In my opinion it is important data that should be published, however I have a few edits/comments worth mentioning.

I have made in-text comments in my attached marked PDF, but I will list below general comments to consider (some of which may be redundant to comments in the PDF!).

- Based on the data presented here, I disagree with the interpretation that form and function in the mandible "progressed" (line 108) in these early synapsids. Especially when considering Fig. S4, it appears that in general, 'pelycosaurs' group together and therapsids group together. Further, significant differences between groups look to me to generally delineate 'pelycosaur'-grade synapsids from therapsids.
- While I do appreciate the difficulties of incorporating teeth into this work and understand the rationale not to include them, it is a critical part of the feeding system and I would urge the authors to discussion or mention this limitation in their discussion.
- I am in favor of speculative discussion points—it is what keeps the field moving forwards—however, I urge caution in the certainty of the language used in the discussion section. I have noted a few problematic sections in the marked PDF, but I would suggest the authors ensure that they are careful

to conclude only on hypotheses tested in this study and to employ more open-ended language on hypotheses generated (many of which I find exciting!) but not tested in this study.

- I believe there is an important discussion point to be made here about how this work fits into the broader and recent research on this transition ('pelycosaur' to therapsid). The authors touch on locomotion as it related to predation, but their findings also support a growing body of evidence of a distinction between basal synapsids and therapsids. (See Lungmus and Angielczk 2019 PNAS & Jones et al. 2018 Science).
- There are few inappropriate or missing citations that I have edited in the marked PDF.
- For the functional feeding subgroups, I would suggest the authors not employ teeth as their icon for each type. Teeth were not incorporated into this study and there is important variation in the size, shape, and heterogeneity of teeth in at least some of these groups that is misrepresented by these symbols.

~Megan Whitney

Reviewers' comments:

Reviewer #1 (Remarks to the Author):

General comments to authors

Your paper is a strong contribution to our understanding of synapsid carnivore evolution. The foundation of the study is morphometric analyses of jaw shape, but you've expanded on those analyses in interesting ways, such as classifying taxa into functional feeding groups, incorporating phylogeny into some analyses, and using the results to infer predator-prey interactions. In some places I think you may be a little too assertive about your inferences of feeding behaviors and ecological interactions – inferring behaviors/ecologies from jaws (and size) alone is a bit speculative (see comments below). But that said, I find the ecology-focused aspect of this paper much more interesting and informative than similar studies that only focus on morphology or disparity patterns. And I think it'll be of interest to a broad readership.

Admittedly, my taxon expertise lies in the crown mammal portion of the synapsid tree, so I don't have any comments on specific taxa or how you classified them into feeding groups, but hopefully other reviewers/editors can evaluate that aspect of the paper. Instead, I focused my review mostly on the data and analyses. My only major concern is about how you treat the functional vs shape data, which I explain below. Most comments are meant to be constructive suggestions rather than critiques.

Feel free to contact me if you have any questions about my review.

Dave Grossnickle
dmgrossn@uw.edu

- Thank you for your positive thoughts on our study and we are grateful for the comments and welcome your perspectives on our work. We have responded in detail below.

Major comments

I strongly support your use of functional traits in morphometric analyses. However, I have concerns with how you present your functional traits and analyses.

First, your 'functional' data and 'shape' data both capture very similar jaw information. The functional traits are measured using shape traits such as jaw length, jaw height, in-levers (measured to jaw margins), tooth row length, etc. (Fig S2), and these traits are all captured (along with other traits) by the geometric morphometric shape analyses (Fig S1). And this overlap is reflected in your results – the multivariate plots/disparity patterns for shape and function look very similar (Figs. 1, 4, 5).

This issue is magnified because we don't know how well your functional traits are tied to functions due to the lack of behavioral data or modern analogs for extinct synapsids (as you acknowledge, e.g. Line 79). Most of your functional traits seem to be chosen based on traits of modern taxa (based on your citations in the Supplementary Methods), but these may not be relevant to early synapsids. For instance, therian mammals chew in a fundamentally different way than earlier synapsids (e.g. Grossnickle et al. 2022 ZJLS), and thus their functional jaw traits may be very unique to those of earlier synapsids. In other words, your functional analyses may not be capturing functional information any better than your shape analyses.

This overlap between ‘function’ and ‘shape’ analyses by itself isn’t necessarily problematic. But my concern is that you present them as two independent analyses (e.g. Lines 88-91) – shape vs function – which seems a bit misleading. And, the analyses feel redundant. Thus, my recommendation is to treat one analysis (maybe shape) as primary and the other analysis (maybe function) as supplemental, and describe BOTH analyses as alternative morpho-functional or ecomorphological analyses (rather than labeling them as ‘shape’ and ‘function’ analyses). This would help to simplify the complex figures. For instance, I don’t think you need stacked morphospace plots (Fig 5) for both ‘shape’ and ‘function’ (they look very similar), and you could move one to the Supplement. I think the most valuable functional results are the univariate plots in Figure 2, which I recommend keeping in the main text.

- We appreciate the reviewer’s comments and have included additional text to mitigate their concerns, but we prefer to keep both analyses within the main text. With regards, to the issue of how we present our analyses and the potential redundancy of including both, we have added the following text in the introduction to more clearly explain the choice of using both morphometric approaches in this study and avoid any confusion on their usage: “Furthermore, through combined use of geometric and traditional, linear measurement-based morphometric methods³¹, we provide a broad assessment that captures carnivorous synapsid jaw morphofunctional evolution from differing perspectives and partially mitigates the divergent impacts of phylogenetic heritage, taxonomic scaling, or methodological choices³⁷.”

We also include the following text in at the beginning of the results section to reinforce this point: “Geometric morphometric landmark data (Supplementary Fig. S1) and standardised functional measurements (SFM) (Supplementary Fig. S2) (See Supplementary Methods) were used to study synapsid jaw evolution following the approach of Singh et al.³¹, as they provide two slightly different but complementary perspectives on jaw morphofunction³⁶. Both approaches are tied to morphology, but the geometric method captures unambiguous differences in shape, whereas the functional measurement approach provides more direct assessment of potential functionality across different jaw morphologies. This approach also provides some assurance that trends and differences shown here are grounded in real patterns of change as non-mammalian synapsid jaw evolution encompasses significant changes in jaw structure, musculature and function³⁷. We henceforth interpret and refer to these analyses as more reflective of jaw form and function, respectively. Furthermore, trends in form and functional evolution are often linked, but do not necessarily correspond³⁶.”

Additionally, having added this clarification to stress that both analyses look at jaw functional morphology from different perspectives favouring either shape or potential function, we believe that relabelling both analyses as suggested would add confusion rather than subtract from it; ‘shape’ and ‘function’ offer more intuitive summary of the different perspectives shown.

Regarding the issue of the redundancy and independence, we agree that some of the characters used in the function PCA such as the relative lengths and ratios, are descriptive of shape, but these characters have been widely interpreted as being related to jaw function across many different vertebrate groups. These characters capture functional aspects of the jaw, such as the points of articulation and muscle attachments, which are not clearly distinguished in the overall shape analysis. The lever measurements used in the mechanical advantage characters relate to distinct points of the jaw that are lost in consideration of the overall shape of the jaw, particularly in non-mammalian synapsids, which saw gradual reorganisation of their jaw musculature through their evolution. As

such, we have added in additional clarification but keep both analyses in the main text for readers to consider as they wish.

With regards to the potential uncertainty of our functional characters, we fully agree that non-mammalian synapsids lack compelling analogues to fully validate our functional interpretations, but we feel that because our chosen characters have been so widely applied in morphometric analyses of fish, reptiles, and mammals (see above), they encompass the phylogenetic bracket of potential jaw functionality seen in non-mammalian synapsids, which sit along a spectrum between more reptilian or mammalian jaws. These characters were also chosen as they represent somewhat basic aspects of jaw morphofunction across a wide range of taxa. For example, reinforcing the symphysis would improve resistance to stresses during biting in the jaw of a mammal or reptile. Similarly, changes to mechanical advantage are going to impact the application of bite forces. We use these basic changes to inform our interpretations of feeding action and do not elaborate on more complex interpretations of jaw mechanics, such as the chewing action cited by the reviewer, as we completely agree that such behaviours cannot be clearly inferred here.

A related issue is that some of your MAMA/MPMA functional traits are based on subjective muscle insertion reconstructions (Fig S2), and I don't see an explanation on how these muscle insertion sites were reconstructed. Did you avoid reconstructing the masseter on delicate reflective laminae/angular bones of many synapsid groups (e.g. see the *Thrinaxodon* reconstruction in Lautenschlager et al. 2017 Biological Reviews)? If you reconstructed the masseter on the angular of some taxa, which ones? Did you utilize reconstructions in previous studies (e.g., Lautenschlager et al. 2017)? These types of details should be explained. I realize that paleontologists must often make subjective decisions about muscle locations (and I'm certainly guilty of it). But, at minimum, you should better explain your decisions, and you should acknowledge and address uncertainties.

- We thank the reviewer for raising this point to us and agree that such details were needed given the uncertainty of jaw muscle reconstructions in non-mammalian synapsids. We have now provided a more detailed account in the Supplementary Methods of the models of jaw musculature we used to reconstruct the locations of the different muscle groups across our different taxa:

“Characters 1–3 are based on using lever mechanics to describe mandibular function, with the jaw acting as a third-order lever system^{2,3}. The adductor musculature provides the input force, the craniomandibular joint acts as a fulcrum, and an output force is produced along the toothrow. Levers are measured from the craniomandibular joint, which is the point of overall jaw articulation, to a point along the jaw margin that is perpendicular to the mid-point of the relevant muscle (Fig. 1). Using margins points provides distinct points that avoid the uncertainty that may arise from trying to identify the exact centre of muscle group attachments across specimens. Taxa with low mechanical advantage (MA) typically exhibit weak, fast bites^{4,5}. Inversely, taxa with high MA possess slower, more powerful bites. In characters one and two, the distance from the jaw adductor muscle attachment to the jaw joint represents the inlever. For characters one and two, we use the mean MA values generated from the MA of each of the three main jaw muscle groups as a more cautious measure of MA, although it should be noted that this inherently reduces the signal of therapsid jaw musculature modifications⁶. These modifications also encompass the earliest stages of jaw anatomy reorganisation that eventually produced the novel mammalian jaw and ear⁷⁻¹⁰.

Reorganisation of the adductor musculature and wider jaw anatomy say shifts in the attachment of the adductor muscles on the lower jaw, resulting in slightly different positions for the inlever tangents required to calculate MA across basal synapsids, non-cynodont therapsids and cynodont therapsids (Fig. 1). For basal synapsids, we largely follow the pattern of adductor muscle attachment laid out by Barghusen¹¹, in which the attachments of the primary adductor muscle groups (adductor mandibulae externus – MAME, posterior – MAMP, and internus pterygoideus - MAMIpt) are not especially derived from that of sauropsids. Therefore, in more basal groups such as varanopids, we defer to attachment reconstructions as used for other sauropsid groups^{8,12,13}. The MAME and MAMP attach to the dorsal, dorsolateral, and dorsomedial margins of the lower jaw, between the coronoid and jaw articulation, while the MAMIpt attached further posteroventrally, to the lower margin of the angular, immediately posterior to the reflected lamina¹¹ (Fig. 1). Jaw modification across the basal synapsid-therapsid transition changes the angle and length of the moment arms^{6-8,10}, but the muscle group attachment sites on the lower jaw generally remain quite similar to as seen in *Dimetrodon*^{11,14}. The expanded reflected lamina present in therapsids required careful consideration of the literature and both the lateral and medial view images of the jaw where available to assess angular morphology and avoid reconstructing muscle insertions on the reflected lamina, although gorgonopsians are thought to have had some attachment on this structure^{15,16}. Thrinaxodontid cynodonts also required a slightly different regime as the adductor musculature attachment began to shift from the postdentary bones onto the dentary in these early cynodonts^{9,14,17}.”

We have also edited Supplementary Figure 2 to include some of this new detail on muscle reconstructions.

The figures are attractive and packed with information. However, they rely heavily on multiple color schemes, which can be confusing. (And they might be hard for colorblind readers to interpret – have you checked that the color choices are appropriate?) For instance, Figure 6 includes color schemes for FFGs (mostly shades of red), geologic epochs (mostly shades of red), and clades, and geologic events are marked by a red circle/line. Where possible, I recommend reducing the number of colors (or color schemes) used. The colors for the geologic ages are probably least important, so maybe you could remove those colors in all the figures? You could replace them with alternating gray and white, like you use in the background of Fig 7.

- We appreciate the reviewer’s concerns and have amended the figures to simplify them where possible such as by following the reviewer’s recommendation to replace the colour coded geological timebins with alternating grey and white bands. All figures have also now been assessed using a colour blindness simulator to check whether they are suitable for colour blind readers.

You use a lot of ‘-ism’ and ‘-ity’ words. Maybe it’s just personal opinion, but I find that style of writing harder to read. For instance, you say “using the well-established ecological utility of mandibular functionality and body size” (Line 78), which didn’t make any sense to me the first time I read it. I’d probably re-phrase that to something like “using jaw functional morphology and body size, which are well-established correlates of ecological traits.” Another example is “growing antagonism and dynamism ... of interactions” and “complexification” (Lines 18-20). I’d instead prefer phrases such as “an increase in antagonistic interactions” and “increased complexity”. But this is just my opinion, so feel free to ignore my comment if you’d like.

- We understand and appreciate the reviewer’s comment and have tried to make changes following their recommendation, which are highlighted throughout the text. Regarding the specific lines highlighted, we use their suggested phrasing for line 78, but not for lines 18-20 as it would take the text beyond the Communications Biology word limit for the abstract.

You use a LOT of acronyms, which are hard for a reader to keep straight and often seems unnecessary. For instance, OE is defined and then only used once (I think) in the main text (although it’s also used in the figures). And abbreviating phrases like ‘morphospace occupation’ as MO seems unnecessary. Having too many acronyms results in phrases like this: “... to optimise MPMA (Figs. 1–2), resurrecting the PBS FFsG” (Line 277), which won’t make sense to a reader who hasn’t memorized the acronyms. I recommend removing any acronyms that aren’t used regularly throughout the paper.

- We understand and appreciate the reviewers comment and have gone through the text to try and reduce our use of acronyms – much of their initial use was to try and be more concise. These changes are highlighted in text.

Much of your later discussion is on ecological inferences, which I like. But an issue is that some topics are challenging to address using a limited sample of taxa that are temporally scattered over tens of millions of years. For instance, you discuss niche partitioning, which can result from the coexistence of competing taxa (Line 341). Maybe I missed it, but I don’t see any discussion on evidence that species in your sample coexisted/interacted. How much temporal and geographical overlap was there among taxa in your sample? Are many of the taxa found at the same fossil localities?

I think it’d be safer to frame your discussion more broadly on ‘ecological/functional diversification’ rather than increased ‘niche partitioning’. Your results demonstrate ecological diversification, whereas niche partitioning is just one potential cause for ecological diversification (along with increased geographical isolation, stochastic change/genetic drift, etc). If you maintain the niche partitioning discussion, then I recommend tempering your comments and maybe presenting your conclusions as hypotheses for future testing.

- We appreciate the reviewer’s comment and accept that we may have been a little too strident when discussing the potential for niche partitioning and additional inferences. We have tried to temper these in text. Regarding niche partitioning, the broader patterns of FFsG prevalence reported here may partially reflect some community-level patterns of trophic niche diversity given that many Palaeozoic synapsid taxa are known from a relatively few localities, so many (particularly therapsid) taxa included in this study likely coexisted. Nonetheless, we decided to more explicitly assess this and have now included an additional plot by assemblage (Figure 8) to properly highlight potential intra-guild dynamics and patterns of niche partitioning. This revealed more complex patterns of ecological differentiation and we have expanded the results and discussion section to reflect this. These new sections are included here:

“**Ecometric patterns within carnivorous synapsid assemblages.** Ecological similarity between multiple, closely related, sympatric lineages can produce strong competitive pressures⁸⁷. Therefore, the close relatedness of different carnivorous synapsids in successive faunas implies that intraguild competition was a powerful selective pressure and potentially a driving force in their diversification through the Palaeozoic. Growing feeding functional group diversity and the emergence of new speed specialist and power shearer subgroups (Figs. 6-7), may signal growing evolutionary pressures on terrestrial carnivores across the Permian. Assemblage-level patterns of FFsG and body size (log₁₀ femur length) distributions enable examination of potential niche overlap through the late

Palaeozoic and earliest Triassic (Fig. 8). It should be noted that these patterns represent a limited snapshot of late Palaeozoic assemblages, selected for their relatively rich carnivore communities, which may not reflect trends across broader geographic and/or temporal ranges.

Despite some declines through the latest Guadalupian and PTME, there is a punctuated expansion of trophic diversity across carnivorous synapsid communities through the late Palaeozoic. The potential for trophic niche overlap appears to have grown through the Permian. Clade-based sorting by FFG and size was strong at an assemblage level, and strongest in the early basal synapsid-dominated assemblages of the El Cobre Canyon and Arroyo⁸⁸ Formations (Fig. 8). The prevalence of taxa sharing FFsG functionalities increased through time and was generally quite high in the therapsid-dominated assemblages of the mid-late Permian. Clade-based size tiering remained fairly strong, except across the Guadalupian-Lopingian transition (Figs. 6b; 8); there was reduced size and FFG differentiation across the ECE and the emergence of theriodont-dominated carnivore assemblages. Stronger size-stratification re-emerged through the Lopingian and restored a pattern seen in earlier assemblages in which raptorial specialists were the smallest carnivores and power shearers are the largest. Russian Lopingian assemblages generally show stronger size tiering than their African counterparts perhaps reflecting regional differences in ecosystem structures or fossil sampling. The *Lystrosaurus declivis* Assemblage Zone⁸⁹ captures the miniaturisation of synapsids following the PTME⁷³ and supports minimal changes to FFsG and relative size distributions from the latest Permian (Figs. 6,8).

Size distributions show that certain clades monopolised large apex predator niches through certain intervals; sphenacodontids in the early Permian, dinocephalians through most of the middle Permian, and gorgonopsians in the late Permian (Fig. 8). Basal synapsid clades converged towards larger sizes through their span as dominant terrestrial carnivores⁸¹, whereas therapsids diversified across both smaller and larger sizes⁹⁰, emphasising body size as a key aspect of therapsid ecological variation (Fig. 8; Supplementary Fig. S8). Niche differentiation in carnivorous therapsids appears primarily driven by varying body size rather than jaw morphofunction (Fig. 8). Body size ranges contracted through Olson's extinction, the ECE, and the PTME in a recurrent pattern of size reductions through intervals of environmental instability in non-mammalian synapsids⁴. Size-based sorting of clades across and within FFsGs also weakened through these events (Figs. 6-8; Supplementary Fig. S8).

The present categorical approach likely underestimates ecomorphological differences as the FFGs mask subtle differences in jaw functional morphology. However, the increasing occurrence of sympatric taxa with shared FFsG functionality and similar body sizes tentatively supports high ecomorphological similarity and so strong potential niche overlap and intra-guild conflict among carnivorous therapsids (Fig. 8).

Potential niche partitioning and intraguild dynamics among carnivorous synapsids.

Sympatric taxa within the same ecological guild are generally expected to engage in niche partitioning, which enables them to minimise competitive pressures and so maximise their feeding efficiency through morphological and/or size differentiation^{24,91}. Yet, coexisting carnivorous synapsids exhibited strong ecomorphological similarity for much of the Permian (Fig. 8). Consequently, their diversification through most of the mid-late Permian appears driven by niche packing rather than niche expansion⁹². The origination of new FF(s)Gs serve as clear examples of niche expansion⁹² that generally coincide with wider extrinsic changes such as the radiation of new prey and climatic events⁹³ (Figs. 6-7). Nonetheless, reduced size differences between sympatric taxa in

successive assemblages suggest subdivision of existing niche space through body size (Fig. 8). Niche packing is linked to high net primary productivity, which supports the abundant resources required to enable the stable coexistence of many ecologically similar taxa and/or more complex ecosystems that permit the increased subdivision of existing niche space^{94,95}. Therefore, Lopingian ecosystems may have been highly productive and heterogenous^{96,97}, given that they hosted such rich communities of ecologically conservative carnivorous therapsids.

Though limited, we find some examples of ecomorphological divergence suggestive of niche partitioning and potentially ecological displacement¹⁹. Ophiacodonts and sphenacodontians diverged from shared forceful gripper jaw functionality as sphenacodontians grew larger and developed more robust jaws through the Carboniferous-Permian transition (Figs. 6-8; Supplementary Fig. S8). Therocephalians and gorgonopsians similarly diverged through the Guadalupian-Lopingian transition from sharing the hypercarnivorous SBS eco-functionality as highlighted by their modified jaw adductor musculature and articulation to support powerful bites and resist disarticulation during prey capture^{54,64,98}. Lopingian gorgonopsians predominantly diversified across hypercarnivorous power shearer niches, although some smaller, more gracile taxa such as *Aelurosaurus felinus* and *Lycaenops ornatus* demonstrate that gorgonopsians maintained a presence within the the speed specialist FFG (Supplementary Data S3, 7). Therocephalians expanded mostly across the speed specialist FFG and into more mesocarnivorous niches (Figs. 4, 6–8; Supplementary Fig. S8b). Therocephalians and gorgonopsians became further stratified by size as they were segregated between FFGs (Figs. 4, 6-8; Supplementary Fig. S8). Size differences between these clades are more pronounced in cases where sympatric taxa belong to the same FFG, further exemplifying size as a key aspect of therapsid niche differentiation (Figs. 6, 8). Theriodont emergence in the middle Permian also precipitated apparent niche adjustments in dinocephalians and biarmosuchians that may have contributed to their respective extinctions. Both clades vacated the newly contested ecospace as dinocephalians became larger, while biarmosuchians became smaller and exclusively speed specialists (Figs. 4, 6–8; Supplementary Fig. S8). Becoming larger would likely have heightened dinocephalian sensitivity to eco-environmental changes and so supported their extinction in the ECE. Whereas biarmosuchian specialisation may have limited their adaptability when therocephalians and gorgonopsians radiated into similar, small speed specialist niches in the latest Capitanian, encouraging the decline in biarmosuchian disparity and FFG prominence (Figs. 4, 6c-8).

Feeding morphology and body size are consistent avenues of variation for niche partitioning through the Phanerozoic^{31,99-101}, but it seems not especially among carnivorous therapsids through much of the mid-late Permian (Fig. 8). Body size appears more significant role for ecological differentiation than jaw morphofunction, which is somewhat understandable given the limited mechanical variation of most prey tissues/materials. However, assemblage-level patterns of size variation among therapsids hint at further alternative means of niche divergence. Minimal ecomorphological differences, particularly in size, between coexisting therapsid carnivores echo patterns among extant mammalian carnivores in rich communities as seen in the African Serengeti¹⁰² rather than other rich non-mammalian predator communities from deep time¹⁰³. These minimal differences have been linked to limited prey escape behaviour and spatial distributions driving particular predator attack behaviour¹⁰². The arid, savannah-woodland mosaic environment of the modern Serengeti hosts a high diversity and abundance of prey, further highlighting the potential richness of some Lopingian ecosystems^{96,102}. However, the rich mammalian carnivore communities found in the

Serengeti and elsewhere also demonstrate complex patterns of antagonistic interference competition^{3,24,25}. Larger mammalian carnivores tend to enjoy less hunting success than smaller competitors but are better kleptoparasites, being more successful in interspecific battles over carcasses^{3,102,104}. This drives smaller sympatric predators to pursue ecological differentiation by adjusting their spatiotemporal foraging preferences^{3,25,102,104,105}. Therefore, heightened jaw morphofunctional similarities offset by increased size differentiation may imply growing behavioural complexity among therapsids through the Permian. Scleral ring and orbit size variation indicate diverse diel activity patterns across non-mammalian synapsids¹⁰⁶ supporting their ability to temporally partition carnivorous niches. Moreover, ecological differentiation could have been primarily realized through postcranial aspects of skeletal anatomy.

Impacts of synapsid postcranial evolution on trophic ecology. Carnivore foraging and prey preferences are closely linked to their speed, agility, and endurance^{104,107}. Whether a predator employs ambush or pursuit hunting techniques across a small or large range of territory is linked to their locomotor capabilities. The forelimbs may also be involved in prey capture, being used to pin down and stabilise prey, thereby easing the stresses and strains on the jaws^{55,66}. Furthermore, by shaping these behaviours, locomotor traits affect how sympatric carnivores vary their habitat and foraging preferences to differentiate their niches and minimise intra-guild conflict^{24,25,104-105,107}.

Axial and appendicular anatomy offer some indication of locomotor capabilities and were key loci of evolutionary change for synapsids through the late Palaeozoic. Synapsid axial and appendicular evolution is linked to advances in locomotor efficiency⁶² and denotes their growing propensity for terrestrial lifestyles. Trends towards reduced lateral undulation during ambulation, increasingly erect postures, forward-facing feet, and increasingly parasagittal gaits point to enhanced stability and efficiency during locomotion^{50,62,108-112}. Remodelling of the pectoral girdle and shoulder joint permitted a wider range of forelimb motion supporting manoeuvrability across rough terrain and wider utility beyond locomotion, despite the forelimbs retaining a sprawling position¹¹³. Greater morphological disparity across the basal synapsid-therapsid transition further demonstrates the increasing utility of the forelimb in Palaeozoic synapsids^{114,115}. Growing locomotor capabilities likely enabled new foraging and prey capture techniques by synapsid carnivores.

Shifting emphasis from catching to killing prey across the basal synapsid-therapsid transition (Figs. 3, 7) may reflect a shift from passive ‘sit and wait’ hunting towards an ‘active search’ approach characterised by seeking out prey¹¹⁶. Active search hunting is linked to greater locomotor efficiency¹¹⁷ and typically employed by larger predators that track prey over distance^{102,116,118,119}. Larger sizes (Figs. 6-8) combined with further auditory¹²⁰⁻¹²² and olfactory modifications¹²³⁻¹²⁵ may have supported capabilities for tracking of prey over longer distances among some therapsid carnivores. Increased energetic constraints on larger carnivores^{126,127} would have driven more antagonistic interference competition, echoing size-based intra-guild relationships in present faunas^{24,104}, creating strong selective pressure for further ecological differentiation via improved locomotory and foraging efficiency¹⁰². Postcranial morphology^{108,109,128} and fossil trackways¹²⁹ point to greater ambulatory speeds among carnivorous (theriodont) therapsids and suggest they were capable of small bursts of acceleration, supporting the idea of felid-like ambush hunting in some carnivorous therapsids¹⁰². Therapsid torso morphology pulls their centre of mass anteriorly relative to basal synapsids, towards the thoracic region and the heavily built, splayed forelimbs, which helps anchor therapsid predators when attacking prey with their jaws¹¹⁶. This

stabilisation further supported lateral head movements, facilitating slashing attacks against prey¹³⁰, and emphasising the offensive utility of the jaws (Fig. 3). Additional emphasis on damaging and so quickly incapacitating prey may also partially reflect the growing mobility of prey when considered alongside tetrapod locomotor evolution through the late Palaeozoic¹¹², as quick kills would have limited opportunities for prey to escape as well as cause injury to the attacking predator.

Patterns of trophic differentiation observed here may relate to differences in locomotory ability as synapsids evolved a growing aptitude for terrestrial lifestyles¹⁵. Differences in postcranial anatomy may indicate that FFG divergence between ophiacodonts and sphenacodontians reflects divergent preferences for semi-aquatic and terrestrial habitats, respectively^{27,131} (Figs. 4, 6; Supplementary Fig. S8). Differences in locomotor capability may have also shaped aforementioned examples of potential therapsid niche partitioning. Theriodonts possessed greater locomotory and appendicular capabilities than biarmosuchians and dinocephalians^{62,102}, which may have supported their survival through the ECE and subsequently allowed them to overtake surviving biarmosuchians (Fig. 6). Advances in locomotor ability are also associated with respiratory and metabolic changes linked to the origin of synapsid endothermy^{8,114,132,133}. Endothermy allows mammals to regulate and maintain high body temperatures independently of external environmental conditions, unlike most reptiles, which are ectothermic. Dating the emergence of synapsid endothermy remains challenging, but it could be that the differentiation of gorgonopsians, therocephalians and cynodonts in the Lopingian (Figs. 6–8) could partially stem from early endothermy in eutheriodonts^{133,134}, as the adaptive flexibility it afforded may have enabled eutheriodonts to diversify across smaller sizes and mesocarnivorous niches, as well as survive the PTME. However, much study is needed to clarify the presence of endothermy across Permian therapsids.

Despite growing similarity to mammals, non-mammalian synapsids still lacked the axial and appendicular flexibility^{50,62,108,111,112} and likely metabolic capabilities^{114,132-134} to pursue or grapple prey exactly as seen in modern mammalian carnivores^{66,67,102}, so drawing exact parallels is problematic. Further detailed study is required to better assess non-mammalian synapsid locomotory functional diversity and properly evaluate the ecological context.

Minor/line comments

Abstract: I don't think you mention jaws in the Abstract. I recommend mentioning jaw shape because 1) it's the foundation of the study and 2) otherwise lines like "specialised for either power or speed" could be confused with non-jaw related traits (e.g. locomotor speed).

- We have now made the following revision to better outline that our study revolves around jaw anatomy: "Using morphometric and phylogenetic comparative methods, we chart carnivorous synapsid trophic morphology from the latest Carboniferous to the earliest Triassic (307-251.2 Ma). We find a major morphofunctional shift in synapsid carnivory between the early and middle Permian, via the addition of new feeding modes increasingly specialised for greater biting power or speed that captures the growing antagonism and dynamism of terrestrial tetrapod predator-prey interactions". We feel that this is sufficient clarification, given that we now mention trophic morphology, and this study also considers body size.

Lines 9-10: The first couple of sentences could be revised for clarity. You might revise the first sentence to start with something like "Terrestrial ecosystems increased in complexity in

the ...”. And in the second sentence I’d expand “among these were” to “among these predators were”; otherwise “these” seems to be referring to “terrestrial ecosystems,” the subject of the first sentence. Also, “than ever” implies that ecosystem complexity was greater in the Permian than at any other time in history, including more recent ecosystems. Is that what you mean?

- We accept the reviewer’s recommendation and have accordingly amended these sentences to: “Terrestrial ecosystems evolved substantially through the Palaeozoic, especially the Permian, gaining much new complexity, especially among predators. Key among these predators were non-mammalian synapsids.” Regarding using “than ever”, we meant that these ecosystems developed more complexity than had been seen previously to that point but now recognise how this may cause confusion. We hope the above changes clarify this.

Line 45: “they sought” implies that the tetrapods consciously tried to evolve certain ways. Their evolution was driven by natural selection, not the lineages themselves. I’d revise the sentence.

- We accept that this wording implies some conscious drive and so have revised “sought” to “evolved” to remove this connotation. The sentence now reads: “Overcoming multiple organismal and environmental constraints, tetrapods became increasingly adept on land as they evolved to better survive and exploit the riches of their new realm”.

Lines 75-85: This is a nice summary/introductory paragraph. But just like in the abstract, you never really introduce that your data is based on jaw shape – you instead say “mandibular functionality” and “jaw functionality,” which is confusing to the reader. Also, you could provide more evidence in this paragraph as to why/how jaw shape is correlated with diet/ecology (maybe from a biomechanical perspective) and thus makes an ideal study system.

- We accept the first point and have made some changes to the text to hopefully minimise any confusion over the anatomy studied here and better link jaw and body size to wider trophic ecology. We do not give more explicit details on how jaw shape may be correlated with diet in the interest of brevity as such detail is found in the results and discussion, and direct explanations of the functional characters are given in the supplementary methods.

“By applying morphometric and macroevolutionary analytical methods including the new consensus clustering method of Singh et al.³¹, we reconstruct and quantitatively classify the feeding ecologies of carnivorous non-mammalian synapsids through the Late Carboniferous – Early Triassic (315.2-251.2 Ma), using jaw functional morphology and body size, both of which closely relate to feeding and foraging behaviour³²⁻³⁵. Through a combination of geometric and traditional linear measurement-based morphometric methods³¹, we provide a broad assessment that captures synapsid jaw morphofunctional evolution from differing perspectives and partially mitigates the divergent impacts of phylogenetic heritage, taxonomic scaling, or methodological choices³⁷.”

Line 88: “mandibular”. I assume based on your jaw images (Figs S1 and S2) that you measured the entire lower jaw (including post-dentary bones), correct? I recommend explicitly stating this in the text. Also, you might consider stating that you use the terms “mandible” or “mandibular” broadly to refer to both the dentary and postdentary bones – some researchers might just associate the mandible with just the dentary and/or only use “mandible” to refer to crown mammal jaws. Alternatively, you could replace “mandible” with “jaw” throughout the paper.

- We accept that our current wording may cause some confusion and thank the reviewer for bringing this distinction to our attention as we were unaware of it. To mitigate any misunderstanding, we have followed their suggestion of using “jaw” instead of “mandible/mandibular” throughout the text.

Line 108-110: The similarities in the morphospaces that you mention here highlights my point about the analyses being very similar and a bit redundant.

- We appreciate the reviewer’s concerns that our use of two analytical streams may be redundant given the similar patterns of the results. We fully accept and highlight these similarities in our results and discussion, but we argue that these results nevertheless differ, and deserve inclusion together because they offer different perspectives on carnivorous synapsid jaw ecomorphology based on geometric and traditional morphometric methods. Our use of both approaches allows for broad examination of patterns of morphological evolution as readers can see how overall shape relates to more specific measurements of particular aspects of jaw shape that are widely considered to have some functional significance. We therefore prefer to leave both in the main text and allow readers the freedom to focus on the results obtained using the methodology that they prefer.

Lines 113-114 (and Fig S3): PC1 and 2 scores are already presented in Figure 1, so I think it’d be more informative to plot PC3 and 4 scores in Figure S3 (rather than 1v3 and 2v3). And rather than calling them “secondary morphospaces,” I’d refer to them more specifically as something like “PC3-PC4 morphospaces” (or “PC1/2-PC3 morphospaces” if you don’t follow my advice).

- We have added an additional morphospace plots featuring PC 3 and 4 to Figure S3 and now refer to all of these plots as ‘additional’ morphospaces rather than ‘secondary’.

Lines 147-194: I like these FFGs, and I like that they’re supported quantitatively by the cluster analysis and LDA. But I was underwhelmed by the references cited and additional empirical evidence to help support and validate these classifications. Maybe there simply hasn’t been much work done on the diets of early carnivores (at least relative to herbivores)? You might double check the literature, including Crompton papers, and DeMar & Barghusen 1971 – I think they tested a hypothesis on the feeding behaviors associated with evolution of the coronoid process. And/or maybe you could expand your discussion on how tooth/skull morphology helps to support the inferred feeding behaviors?

- We have expanded our discussion of the FFGs to provide more detail on how these classifications relate with previous literature across the discussion (- see section “Synapsid carnivore feeding functional groups”). We have included some discussion on how well these results link to cranial & tooth morphology, but we have additional studies in progress that focus in detail on the links between tooth and jaw anatomy and potential ecological inferences. Therefore, we would prefer to leave any more detailed discussion for these future studies.

Line 198: I recommend explaining “kinetic inertial biting system”.

- We accept this recommendation and have included the following text:
 “Non-mammalian synapsids possessed a kinetic inertial biting system in which the jaw muscles were most active when the jaw was open, utilising the jaw’s mass and velocity to maximise bite force during closure^{37,61}.”

Lines 471-475: This text feels a little out-of-place in the Methods. You could move it to the Introduction (maybe the last paragraph, Lines 75-85) or Results.

- We appreciate the reviewer's thoughts on this and have followed their suggestion of moving this text and incorporating it into the introduction as the following: "Furthermore, through combined use of geometric and functional morphometric methods³¹, we provide a broad assessment of carnivorous synapsid ecomorphological evolution that partially mitigates the divergent impacts of phylogenetic heritage, taxonomic scaling, or methodological choices³⁷".

Line 494: Was "areas of muscle attachment" used as a functional trait? I don't think so, unless I misunderstand one of your functional traits? You use muscle inlevers for the MAMA/MPMA traits, but those aren't necessarily capturing areas of muscle attachment. For instance, a small muscle with an anteriorly positioned insertion site would have a small attachment area and a long inlever.

- We agree with the reviewer and have now amended this line to the following to avoid any confusion: "These measurements capture aspects of jaw functionality from the overall jaw shape, such as the lever-arm mechanics of the jaw musculature, articulation, and robusticity, which have been used to characterise overall jaw function and interpret feeding ecology^{32,33,107} (See Supplementary Methods)".

Line 503: What is the "D" in "SFMD"? I don't remember that being defined. And SFM is another example of an acronym that seems unnecessary.

- This was a typo stemming from previous usage as "standardised functional measurement data" and has now been corrected to "SFM", we have kept the use of this acronym as it is used frequently in text but accept the reviewer's broader point of limiting acronyms and have made changes to limit these elsewhere such as in the removal of "GM" and "FM" when referring to geometric and functional methods.

Line 523: "To identify the niches of synapsid carnivores ..." seems vague and potentially misleading. I'd instead say "To classify synapsid carnivores into functional feeding groups ...".

- We accept this point and have followed the reviewer's suggestion, including their suggested phrasing so the line now reads: "To classify carnivorous synapsids into feeding functional groups based directly on their ecomorphology, we use the consensus clustering approach of Singh et al.³¹".

Line 530: Can you use the geometric morphometric landmark data (or Procrustes or PC scores) as input in the cluster analysis? As I explain above, I'm skeptical that the functional data capture function any better than the shape data, so you might try analyzing the shape data and see if results are congruent with the functional data.

- We agree that our geometric and functional analyses both stem from overall jaw shape, but this was an intentional effort to support our exploration of the patterns of synapsid jaw evolution by providing slightly different perspectives on ecomorphology. As mentioned in our response above and highlighted in our supplementary methods, the chosen functional characters capture variation in some important aspects of jaw anatomy that are not well represented by jaw shape, such as the point of articulation and muscle attachments/lever mechanics. These cannot be clearly distinguished by the overall shape analysis, so using the shape data for the clustering analyses would minimise the potential functional differences and instead, maximise the phylogenetic signal, pulling out only

broad shape similarities. Therefore, we prefer to use the functional data for this part of our study.

Line 542: You might also list the LDA's classifications, maybe in your Supplementary Dataset file.

- We accept the reviewer's recommendation and have now included the LDA classifications in the supplementary data excel file as the new S8 (sheet 8).

Figure 1b: I only see these plots mentioned once in the text (Line 97), and the results can't be easily interpreted by readers, so it's unclear to me how they're important for the paper. You might either move them to the Supplement, or, if you think they're important, then add more discussion on them.

- We prefer to keep these plots in the main text as we feel that these plots help to clarify how discussion throughout the text on how patterns of shape change relate to broader functional differences and potential differences in feeding ecology. We now make more discrete reference to these plots across the results and discussion.

Supplementary Materials Lines 139-155: This is a nice summary, but it seems to be missing some details. I assume "MA" is mechanical advantage and is measured as the inlever divided by outlever, which is implied later, but you should explicitly state that. The MAMA and MPMA are means of multiple MA ratios, correct? I don't think you explain that here and don't mention what ratios are averaged (at least that I see), although I think I figured it out from the text in Fig S2. You should provide a little more information about which inlevers correspond to which muscles, and how you determined the inlever positions (see my comments above). In Fig S2 it looks like they're measured to the midpoints of the muscle attachment areas (but to the jaw margins and not to the actual center of the muscle areas), which is fine, but I personally would have probably measured them as perpendicular to the inferred muscle force vector. And as I mention above, the masseter position could be especially problematic because of the postdentary bones in the same region.

- We thank the reviewer for pointing out these missing details and have now added in more detail to better explain our methods. Please see our response to this comment above in the main comments section.

Figure S2: For the X/Y/Z lines, there are some abbreviations (for muscles?) that should be defined (unless I missed it): MPT, MAME, MAMP.

- We thank the reviewer for pointing out these missing definitions for this figure and have now included them in the figure legend.

Supplementary dataset: I appreciate the thorough, well-organized dataset! I only see it cited on Line 498 and in the Data Availability section. It contains valuable information that readers may want to check out, so I recommend citing it where relevant, such as after mentioning femur length data (Line 445) and functional trait data (Line 493).

- We thank the reviewer for their kind comment and accept their recommendation of more heavily citing the supplementary data throughout the text, including the instances specifically mentioned.

Reviewer #2 (Remarks to the Author):

Based on the morphometric and macroevolutionary analysis on the lower jaws of carnivorous synapsids, this paper give a new point on the ecological evolution of terrestrial tetrapod ecosystems. As far as I know, it collected the abundant data (although there are errors), I am not quite familiar with some methods applied in this paper, and I do not repeat the analysis, but as far as I know, the major conclusion is convince. It is worth to be published after revision of the following places.

- We thank the reviewer for their time and consideration of our manuscript and are glad that they find our work to be worthy of publication at Communications Biology. Our responses to their specific comments are presented below.

For Permian, early, middle or late are informal and should be lower case; also for lower and upper.

- This has now been rectified and the resulting changes are highlighted throughout the revised text.

L110 Basal synapsids and therapsids are distinguished principally by reductions in RTL and increasing prominence of the postdentary bones in therapsids.

The postdentary bones are not prominent in some therapsids than ‘pelycosaurs’. The most prominent feature is the reflected lamina. This also different what you stated later (Basal synapsids developed increasing robusticity and enlargement of the mandibular body from varanopids and ophiacodonts to (non-therapsid) sphenacodontians).

- We thank the reviewer for pointing out this typo, and we have now rectified this in text: “Basal synapsids and therapsids are distinguished principally by reductions in relative toothrow length and prominence of the postdentary bones in therapsids.”

L252 The earliest known therapsids in the Roadian possessed fairly robust mandibles that lacked prominent coronoid processes, placing these taxa within the central regions of both morphospaces, adjacent to earlier sphenacodontians.

The only Rodian therapsid in your list is Phthinosaurus, which has a prominent coronoid process. Please check its morphospace!

- We agree with this comment and admit that the statement regarding the coronoid process being less prominent is confusing as we meant it to be in comparison to the later therapsids, so we have now amended the text to better clarify our point: “The earliest known therapsids possessed fairly robust jaws that featured more emergent coronoid processes, placing these taxa within the central regions of both morphospaces, adjacent to earlier sphenacodontians (Figs. 1, 4).”

L431 For the age, you should use the age on the latest version (2022) and change your matrix and figures.

- We have now updated all the first and last appearance dates in our data and results according to the latest (2023) ICS dates.

Reviewer #3 (Remarks to the Author):

Singh et al. employ geometric morphometrics and phylogenetic comparative methods to study the evolution and occupation of different feeding niches in carnivorous basal and therapsid synapsids. They find evidence for discrimination between basal synapsid and

therapsid mandibular form and function by examining a suite of functional measurements. They find increased niche partitioning within carnivorous therapsids and suggest that this disparity in function allowed for increased predator-prey interactions and may have played a role in mass extinction selection barriers and synapsid evolution more broadly. This a well-written paper that is thorough and thoughtful. In my opinion it is important data that should be published, however I have a few edits/comments worth mentioning.

I have made in-text comments in my attached marked PDF, but I will list below general comments to consider (some of which may be redundant to comments in the PDF!).

- We are very pleased the reviewer found our paper interesting and was excited by some of the ideas we put forth. We thank them for their comments and are glad to have their thoughts on how to improve our work. We have responded in detail below:

Based on the data presented here, I disagree with the interpretation that form and function in the mandible “progressed” (line 108) in these early synapsids. Especially when considering Fig. S4, it appears that in general, ‘pelycosaurs’ group together and therapsids group together. Further, significant differences between groups look to me to generally delineate ‘pelycosaur’-grade synapsids from therapsids.

- We agree with the reviewer and believe there has been some confusion on this point due to our phrasing as we did not mean to imply any progression in terms of evolution from basal synapsids to therapsids. Rather, we meant that taxa are arranged in parallel progressions from gracile to robust forms across the morphospaces of both basal synapsids and therapsids. We now recognise this was unclear and have amended the text to better convey our intent:
“Both the form and functional morphospaces show parallel distributions in both basal synapsids and therapsids from relatively gracile, elongate jaws towards more robust morphologies capable of more powerful bites (Fig. 1)”.

While I do appreciate the difficulties of incorporating teeth into this work and understand the rationale not to include them, it is a critical part of the feeding system and I would urge the authors to discuss or mention this limitation in their discussion.

- We agree with the reviewer on the importance of teeth in the vertebrate feeding system and have now included more consideration of the dentition within the discussion of the FFsGs. These passages are as follows:
 - **“Raptorial specialists:** This group is united by their gracile, longirostrine jaws and lengthy tooththrows, and subdivided by differences in robusticity and biting efficiency into the ‘gracile and forceful grippers’ (GG and FG) subgroups (Fig. 2; Supplementary Figs. S5–6a). This group is almost exclusively populated by basal synapsids but includes some biarmosuchian therapsids. Varanopids and ophiacodonts comprise the majority of GG, and larger robust members of both clades as well as most sphenacodontians form the FG subgroup (Fig. 2; Supplementary Fig. S7). Extended tooththrows enable a wide distribution of bite force and suggests an emphasis on gripping and retaining prey, particularly when combined with the conodont teeth present in basal synapsids (Figs. 2–3; **Supplementary Fig. S6**). Their gracile jaws (particularly of the GG) are ill-suited to high stresses associated with comparatively large prey (Fig. 2), suggesting a preference for much smaller, less combative prey such as insects, fish, and smaller tetrapods (Fig. 3). Differences between GG and FG raptorial specialists in MA, areas of muscle attachment, and dentary robusticity (Fig 1b, 4; Supplementary Fig. S6) (Fig. 2) illustrate FG optimisation for biting efficiency and power over biting speed²⁸. Growing tooth size and

shape variation through basal synapsid evolution⁴² supports a shift towards more complex jaw use and feeding behaviour associated with tetrapod-on-tetrapod predation. Raptorial specialist dentition encompasses simplistic conodont to derived ziphodont tooth morphologies, with the ziphodont teeth appearing more commonly in sphenacodontids^{42,43}, illustrating their growing efficiency for shearing flesh and specialisation as tetrapod predators⁴⁴. Raptorial specialists somewhat echo the jaw functionality of some sauropsid reptiles, and such similarity may extend to prey capture/killing behaviour (Fig. 3). Varanopids were well suited to rapid head movements⁴⁵ like modern varanid lizards, which employ such movements when grasping and killing prey^{37,46}. The basal phylogenetic position of varanopids indicates such behaviour may be plesiomorphic for synapsid predators. However, varanid prey capture and dismemberment heavily involve their neck and forelimbs^{47,48}, whereas basal synapsids likely relied primarily on their jaws to manipulate prey.

- **Speed specialists:** All therapsid clades, particularly therocephalians, are represented within this group, which is subdivided into ‘grip and rip attackers’ (GRA) and ‘rapid light attackers’ (RLA) (Fig. 2; Supplementary Fig. S6). Much like the raptorial specialists, speed specialists exhibit low MA indicative of fast bite speeds³³, moderate robusticity, and prominently curved dentaries to improve their grip on prey (Fig. 2; Supplementary Figs. S5–6). RLA speed specialists modified these traits further, in addition to reducing their OMA to further enhance bite speed (Fig. 2). However, speed specialists show more limited distribution of biting force towards the front of the jaw, as typically illustrated by a shorter tooththrow and reinforced symphysis. Such modifications demonstrate enhanced focus on the penetrative and gripping power of the jaws. RTL is generally shorter than in raptorial specialists but therapsid speed specialists still possess lengthier tooththrows than seen in therapsid power shearers (Fig. 1b, 2; Supplementary Fig. S6). These tooththrows generally feature highly enlarged ‘pre-canine’ and ‘canine’ teeth, and smaller but often more complex post-canine teeth⁴². Greater emphasis on anterior biting efficiency boosted the penetrative power of the anterior dentition, while greater post-canine complexity combined with increased jaw robusticity further enhanced the ability to hold and resist struggling prey. Alongside higher biting speeds, these features suggest that these therapsids potentially employed a ‘harrying’ manner of prey capture, like that seen in canids⁴⁹ (Fig. 3). Such behaviour is somewhat consistent with jaw and neck muscle development across the basal synapsid-therapsid transition, although therapsids lacked the axial flexibility and speed⁵⁰ to fully replicate canid hunting, which often involves extended pursuits⁴⁹. While better suited than the raptorial specialists to extended struggles with prey, the elongate jaws of GRAs remain unsuited to the higher stresses associated with such struggles with large prey. Therefore, GRAs likely targeted relatively smaller prey; using their own larger size to inflict disproportionately more damage and quickly subdue prey⁵¹ (Figs. 2–3, 7; Supplementary Fig. S7). More robust forms could perhaps tackle larger, more similarly sized prey.
- **Power shearers:** This group mostly comprises gorgonopsians and basal therocephalians, but also features most cynodonts and multiple species of *Dimetrodon* and *Sphenacodon* within the power bite specialist (PBS) subgroup (Figs. 2–3; Supplementary Figs. S5–6a). This group is distinguished by highly robust and mechanically efficient jaws with substantial symphyseal reinforcement. While shearing bite specialists (SBS) form the core of the group, power bite specialists (PBS) and deep shearing specialists (DSS) represent distinct variations on the power shearer feeding mode, opting to maximise biting efficiency along the entirety or just the anterior margin of the tooththrow, respectively (Fig. 2; Supplementary Figs. S6). The DSS are dominated by rubidgeine gorgonopsids, which were the largest and most heavily built gorgonopsians⁵². SBS and DSS appear optimised for penetrating deeply into and shearing off chunks of prey tissue, supporting previous inferences of hypercarnivorous behaviour derived from highly incisiform anterior dentition^{42,53} and cranial morphology^{53,54}. Hypercarnivory is also

evident from the prevalence of hypertrophied canines across gorgonopsians and therocephalians³⁷, and the symphyseal reinforcement exhibited across this FFG likely also evolved to support these teeth against lateral stresses as in sabertoothed cats⁵⁵. In contrast, the PBS exhibit greater mean posterior MA, suggesting additional emphasis on compressive forces during feeding (Figs. 2–3). Nonetheless, differences in size (Supplementary Figs. S6–7), dentition⁴² and musculature²⁹ within the PBS point to divergent ecologies, with hypercarnivory in the sphenacodontids and dinocephalians, and mesocarnivory or omnivory in cynodonts. Sphenacodontid PBS possessed large ziphodont and/or conodont anterior teeth, and more bulbous posterior teeth⁴² featuring subrounded to ovoid cross sections that afforded high structural strength³⁷. This differentiation indicates dual emphasis on removing flesh and inflicting compressive damage to incapacitate prey rather than acute durophagous feeding behaviour⁴³. Relatively high posterior MA and jaw robusticity (Fig. 1b) make these sphenacodontids well suited to grappling with comparably sized prey^{43,56,57}. Similar emphasis on posterior biting efficiency is also observed in the dinocephalian PBS, *Anteosaurus magnificus*^{37,59} (Supplementary Data S8). The more complex multi-cusped teeth seen in cynodont PBS are associated with enhanced comminution⁶⁰ and suggestive of more durophagous diets.”

I am in favor of speculative discussion points—it is what keeps the field moving forwards—however, I urge caution in the certainty of the language used in the discussion section. I have noted a few problematic sections in the marked PDF, but I would suggest the authors ensure that they are careful to conclude only on hypotheses tested in this study and to employ more open-ended language on hypotheses generated (many of which I find exciting!) but not tested in this study.

- We are glad the reviewer found our points of discussion exciting and perhaps in our own excitement, we were too strong in voicing some of our thoughts stemming from these results. We accept this point and have tried to amend these points through the discussion particularly in the latter parts.

I believe there is an important discussion point to be made here about how this work fits into the broader and recent research on this transition (‘pelycosaur’ to therapsid). The authors touch on locomotion as it related to predation, but their findings also support a growing body of evidence of a distinction between basal synapsids and therapsids. (See Lungmus and Angielczk 2019 PNAS & Jones et al. 2018 Science).

- We agree that this work provides some ecological context for the pelycosaur-therapsid transition and appreciate their suggestion to discuss this transition more broadly. We include more detail in response to the comment below regarding locomotion.

For the functional feeding subgroups, I would suggest the authors not employ teeth as their icon for each type. Teeth were not incorporated into this study and there is important variation in the size, shape, and heterogeneity of teeth in at least some of these groups that is misrepresented by these symbols.

- We understand the reviewer’s comment and now see that our original icons could be misleading, therefore we have removed the toothy icons in all figures.

There are few inappropriate or missing citations that I have edited in the marked PDF.

- We thank the reviewer for their comments and appreciate the annotations on the manuscript pdf as we found them were very helpful from a logistical perspective when going through the revisions for this manuscript. We have included all these comments and our responses below.

Page 2, Line 69: I would recommend adding (or replacing with) Day et al. 2015 as it's more recent and relevant to the mid-Permian extinction. Citation: Day et al. 2015. When and how did the terrestrial mid-Permian mass extinction occur? Evidence from the tetrapod record of the Karoo Basin, South Africa. *ProcB* 282: 20150834.

- We thank the reviewer for their suggestion and have changed the citation to Day et al. as recommended.

Page 4, Line 108: I disagree with this interpretation of the results. Based on Fig. 1, and especially Fig. S4, there appears to be more of a shift between 'pelycosaurs' and therapsids. I don't see a lot of evidence supporting "steps" between clades, but rather a distinction between form and function in basal synapsids and therapsids.

- This comment was included as one of the main comments above and we provide a detailed response there, but for clarity, this point has now been changed to read: "Both the form and functional morphospaces show parallel distributions in basal synapsids and therapsids from relatively gracile, elongate jaws towards more robust morphologies capable of more powerful bites (Fig. 1)".

Page 4, Line 111: I believe it is widely accepted that the postdentary bones become gradually reduced in therapsids. See Sidor 2003 (Evolutionary trends and the origin of the mammalian lower jaw) for reference.

- We completely agree with the reviewer on this point. This was an unfortunate typo and was meant to read "decreasing" not "increasing". We have now corrected this in text.

Page 4, Line 124: I think it is important to note that there is tremendous size variation in this group including much smaller taxa than recorded in these data. I don't think it is entirely necessary to incorporate smaller gorgonopsian taxa here, but just some good food for thought if trends in form and function change between size classes within this group!

- We agree with this point and provide some further detail later in the text to tease apart the similarities and differences between the clades across feeding functional groups and size classes through the discussion, particularly in the section titled, "Potential niche partitioning and intraguild dynamics among carnivorous synapsids".

An additional note on gorgonopsians: the saber-canines (both upper and lower) are huge and it is entirely possible that the massive RSL recovered for especially the large groups included in this study may be equally related to supporting the weight and function of those massive teeth. In all likelihood a large RSL probably serves multiple functions, and I think it's important to reiterate in the text that these traits could have served multiple purposes or evolved for various number of reasons other than jaw function alone.

- We thank the reviewer for pointing this out and have included reference to this in the section describing the power shearer FFG:
"Hypercarnivorous ecologies are also evident from the prevalence of hypertrophied canines across the gorgonopsians and therocephalians³⁷ in these groups, and it is likely that the symphyseal reinforcement exhibited across this FFG also evolved to support these teeth against lateral stresses as in sabertoothed cats⁵⁵."

Page 9, Line 315: While this may also be true, therapsid evolution has been linked to a few other transformations/shifts in their anatomy (and presumably functionality). This has happened in the forelimb (Lungmus and Angielczyk 2019) and the axial skeleton (Jones et al. 2018). I think that it is important to include these papers in a discussion of therapsid evolution especially as it relates to the distinction between basal synapsids and therapsids.

- We completely agree and have clarified this at this point in the discussion so the start of this paragraph now reads:

“Therapsid jaw evolution appears closely linked to increasing predator-prey antagonism”.

While we touched on how patterns of jaw and size evolution reported here may correspond with broader postcranial changes across therapsid anatomy, we accept the reviewers recommendation to expand on these links, and so have now created a new section on this in the discussion:

“Impacts of synapsid postcranial evolution on trophic ecology. Carnivore foraging and prey preferences are closely linked to their speed, agility, and endurance^{104,107}.

Whether a predator employs ambush or pursuit hunting techniques across a small or large range of territory is linked to their locomotor capabilities. The forelimbs may also be involved in prey capture, being used to pin down and stabilise prey, thereby easing the stresses and strains on the jaws^{55,66}. Furthermore, by shaping these behaviours, locomotor traits affect how sympatric carnivores vary their habitat and foraging preferences to differentiate their niches and minimise intra-guild conflict^{24,25,104-105,107}.

Axial and appendicular anatomy offer some indication of locomotor capabilities and were key loci of evolutionary change for synapsids through the late Palaeozoic. Synapsid axial and appendicular evolution is linked to advances in locomotor efficiency⁶² and denotes their growing propensity for terrestrial lifestyles. Trends towards reduced lateral undulation during ambulation, increasingly erect postures, forward-facing feet, and increasingly parasagittal gaits point to enhanced stability and efficiency during locomotion^{50,62,108-112}. Remodelling of the pectoral girdle and shoulder joint permitted a wider range of forelimb motion supporting manoeuvrability across rough terrain and wider utility beyond locomotion, despite the forelimbs retaining a sprawling position¹¹³. Greater morphological disparity across the basal synapsid-therapsid transition further demonstrates the increasing utility of the forelimb in Palaeozoic synapsids^{114,115}. Growing locomotor capabilities likely enabled new foraging and prey capture techniques by synapsid carnivores.

Shifting emphasis from catching to killing prey across the basal synapsid-therapsid transition (Figs. 3, 7) may reflect a shift from passive ‘sit and wait’ hunting towards an ‘active search’ approach characterised by seeking out prey¹¹⁶. Active search hunting is linked to greater locomotory efficiency¹¹⁷ and typically employed by larger predators that track prey over distance^{102,116,118,119}. Larger sizes (Figs. 6-8) combined with further auditory¹²⁰⁻¹²² and olfactory modifications¹²³⁻¹²⁵ may have supported capabilities for tracking of prey over longer distances among some therapsid carnivores. Increased energetic constraints on larger carnivores^{126,127} would have driven more antagonistic interference competition, echoing size-based intra-guild relationships in present faunas^{24,104}, creating strong selective pressure for further ecological differentiation via improved locomotory and foraging efficiency¹⁰². Postcranial morphology^{108,109,128} and fossil trackways¹²⁹ point to greater ambulatory speeds among carnivorous (theriodont) therapsids and suggest they were capable of small bursts of acceleration, supporting the idea of felid-like ambush hunting in some carnivorous therapsids¹⁰². Therapsid torso morphology pulls their centre of mass anteriorly relative to basal synapsids, towards the thoracic region and the heavily built, splayed forelimbs, which helps anchor therapsid predators when attacking prey with their jaws¹¹⁶. This stabilisation further supported lateral head movements, facilitating slashing attacks against prey¹³⁰, and emphasising the offensive utility of the jaws (Fig. 3). Additional emphasis on damaging and so quickly incapacitating prey may also partially reflect the growing mobility of prey when considered alongside tetrapod locomotor evolution

through the late Palaeozoic¹¹², as quick kills would have limited opportunities for prey to escape as well as cause injury to the attacking predator.

Patterns of trophic differentiation observed here may relate to differences in locomotory ability as synapsids evolved a growing aptitude for terrestrial lifestyles¹⁵. Differences in postcranial anatomy may indicate that FFG divergence between ophiacodonts and sphenacodontians reflects divergent preferences for semi-aquatic and terrestrial habitats, respectively^{27,131} (Figs. 4, 6; Supplementary Fig. S8). Differences in locomotor capability may have also shaped aforementioned examples of potential therapsid niche partitioning. Theriodonts possessed greater locomotory and appendicular capabilities than biarmosuchians and dinocephalians^{62,102}, which may have supported their survival through the ECE and subsequently allowed them to overtake surviving biarmosuchians (Fig. 6). Advances in locomotor ability are also associated with respiratory and metabolic changes linked to the origin of synapsid endothermy^{8,114,132,133}. Endothermy allows mammals to regulate and maintain high body temperatures independently of external environmental conditions, unlike most reptiles, which are ectothermic. Dating the emergence of synapsid endothermy remains challenging, but it could be that the differentiation of gorgonopsians, therocephalians and cynodonts in the Lopingian (Figs. 6–8) could partially stem from early endothermy in eutheriodonts^{133,134}, as the adaptive flexibility it afforded may have enabled eutheriodonts to diversify across smaller sizes and mesocarnivorous niches, as well as survive the PTME. However, much study is needed to clarify the presence of endothermy across Permian therapsids.

Despite growing similarity to mammals, non-mammalian synapsids still lacked the axial and appendicular flexibility^{50,62,108,111,112} and likely metabolic capabilities^{114,132-134} to pursue or grapple prey exactly as seen in modern mammalian carnivores^{66,67,102}, so drawing exact parallels is problematic. Further detailed study is required to better assess non-mammalian synapsid locomotory functional diversity and properly evaluate the ecological context.”

Page 10, Line 333: Hopson describes gorgonopsian tooth replacement as more nuanced in this citation rather than simply as rapid replacement. This citation is probably not the most appropriate as it didn't directly examine gorgonopsian replacement but rather, compared with Kermack 1956. I would suggest replacing the Hopson citation with the Kermack citation. Importantly, however, Kermack actually found significantly reduced replacement in gorgonopsian canines especially which is important as they're the primary functional teeth in a gorgon tooth row. I would not conclude that replacement, as it's been studied in gorgonopsians, reflects trends towards hypercarnivory necessarily. But rather, that the large size of their canines make replacement facultatively slow. Therefore, it may not be the most relevant point to highlight in this study.

- We thank the reviewer for pointing us to the Kermack paper and have followed their suggestion to revise this point as follows:
“Gorgonopsians also developed differential patterns of tooth replacement across their toothrow to maintain shearing efficacy, and propalinal jaw articulation to enable wider gapes^{53,54,84}.”

Page 10, Line 352: I am in support of these speculative discussion points and I think they are one-hundred percent a way to keep pushing ideas. However, I would urge the authors to re-read and assess statements such as this one that portray these as tested ideas rather than newly generated hypotheses.

- We accept that we may have been overly forthright in our discussion and have now amended this point as follows to clarify the speculative nature of this point:

“Both responses may have ultimately played a role in the fates of these two clades. Becoming larger would have heightened dinocephalian sensitivity to eco-environmental changes and so contributed to their extinction in the ECE, whereas biarmosuchian specialisation may have limited their ability to adapt when therocephalians and gorgonopsians radiated into smaller speed specialist roles in the latest Capitanian. Indeed, biarmosuchians subsequently declined in morphological disparity and prominence within the speed specialist FFG following this theriodont radiation (Figs. 4, 6c-7)”.

Page 11, Line 373: Again, I would urge caution in the language used as to not conflate what was tested here with discussion points that are important, but not directly tested in this paper.

- We accept that we may have been too strident in our language when discussing potential inferences of our results and have tempered such statements throughout the text. This particular point has been removed.

Page 12, Line 424: A critical caveat to these analyses is the lack of tooth morphology or function into the interpretation of the results. Such work is outside of the scope of this paper, but I believe given the broad readership of this journal, it's important to address this caveat at some point in the discussion.

- We agree that tooth morphology is critical to understanding feeding ecology and have included additional consideration of dentition throughout the discussion, particularly in the section of discussion detailing the different functional feeding group (see section titled, “Synapsid carnivore feeding functional groups”).

However, we must mention that we currently have additional studies in prep that looks specifically at dental variation and potential ecological implications of this variation in non-mammalian synapsids, so would prefer to limit any further discussion of dentition here. We hope the reviewer finds this upcoming work of interest!

Page 27, Figure 2: I would strongly recommend not using teeth as icons for these subgroupings. This paper did not study teeth and within these groups there is variation in tooth shape and size. The use of these icons may be misleading about the kind of analyses completed here.

- This comment was included as one of the main comments above and we give a more detailed response there, but for clarity, we accept this comment and have now changed the icons in all the figures.

Page 30, Figure 5: Please double check that these align with the FAD and LAD data submitted. As far as I know therocephalians FAD is around 270 mya at the earliest and I believe the submitted data have the earliest therocephalian FAD at 268 mya. If I'm correct, then I believe there is an issue with the dark brown THR line in the Eutheriodonts bin.

- We use phylogenetic time-slicing to calculate disparity through time, so these results are influenced by our tree topology and chosen time-bins; we split our stages equally in half to generate an upper and lower substage time-bins. This detail has also now been added into the methods section: “Our time-bins were generated by equally dividing each stage into an upper and lower substage”. We believe this detail may account for some confusion over the disparity results presented. We were unable to calculate upper and lower Roadian SOV values despite incorporation of the phylogenetic inference, likely due our higher resolution timebins and low sampling of early therocephalian diversity. We have amended the figure to better reflect this and we hope the reviewer finds this

consistent with the Radian FAD of therocephalians given by the reviewer and in our supplementary data.

~Megan Whitney

REVIEWERS' COMMENTS:

Reviewer #1 (Remarks to the Author):

Thanks for addressing all my comments in detail. I support publication of your current draft. Congrats on putting together a very nice paper. I have a couple of follow-up thoughts, but you can choose whether or not to follow my suggestions.

I understand your reasoning for wanting to keep both the 'shape' and 'function' analyses, and I like your revisions to the text concerning the similarities of the analyses. Because both the 'shape' and 'function' analyses are variations of similar morphofunctional analyses, you might add parenthetical clarifications after you introduce the two types of analyses -- e.g., "(hereafter 'shape' analyses)" and "(hereafter 'function' analyses)". And you might put single quotation marks around 'shape' and 'function' throughout the manuscript.

One reason why I recommended moving one set of analyses (shape or function) to the supplement is to help simplify the paper and make it more reader-friendly. You have 8 data-packed figures that are a bit overwhelming at first (especially considering that most academics who "read" your paper will simply skim over the abstract and figures). At least in my opinion, the simpler you can make the figures and get your point across, the better. If there are any results that are redundant or non-critical to your conclusions, I recommend moving them to the supplement. But, again, this may just be my opinion, so feel free to ignore my suggestion.

Dave Grossnickle
david.grossnickle@oit.edu

Reviewer #3 (Remarks to the Author):

It is evident from the updated manuscript that Singh et al. have put in considerable effort addressing the reviewers' comments. I am pleased with the edits and additions the authors have made, especially to the discussion sections, and believe this manuscript delivers compelling and interesting research.

REVIEWERS' COMMENTS:

Reviewer #1 (Remarks to the Author):

Thanks for addressing all my comments in detail. I support publication of your current draft. Congrats on putting together a very nice paper. I have a couple of follow-up thoughts, but you can choose whether or not to follow my suggestions.

- We really appreciate the kind words on the revised manuscript and are glad to have your support for publication. You have our thanks for your detailed comments and suggestions, which were incredibly helpful and enabled us to make significant improvements to our paper.

I understand your reasoning for wanting to keep both the 'shape' and 'function' analyses, and I like your revisions to the text concerning the similarities of the analyses. Because both the 'shape' and 'function' analyses are variations of similar morphofunctional analyses, you might add parenthetical clarifications after you introduce the two types of analyses -- e.g., "(hereafter 'shape' analyses)" and "(hereafter 'function' analyses)". And you might put single quotation marks around 'shape' and 'function' throughout the manuscript.

- We accept your suggestion to provide further clarification when introducing the two types of analyses. The text now reads, "We henceforth interpret and refer to these analyses as more reflective of jaw form and function, labelling them as the 'shape' and 'function' analyses, respectively".

One reason why I recommended moving one set of analyses (shape or function) to the supplement is to help simplify the paper and make it more reader-friendly. You have 8 data-packed figures that are a bit overwhelming at first (especially considering that most academics who "read" your paper will simply skim over the abstract and figures). At least in my opinion, the simpler you can make the figures and get your point across, the better. If there are any results that are redundant or non-critical to your conclusions, I recommend moving them to the supplement. But, again, this may just be my opinion, so feel free to ignore my suggestion.

- We very much appreciate your suggestion and understand your view, and robustly agree on making a paper as reader friendly as possible. However, we think that including both sets of analyses in the main text provides firm grounding for the later, more derived analyses regarding the functional feeding groups. Consequently, while we accept that our figures are data-packed, we have strived to make them as clear and intuitive as possible, and hope that readers will appreciate these efforts.

Dave Grossnickle
david.grossnickle@oit.edu

Reviewer #3 (Remarks to the Author):

It is evident from the updated manuscript that Singh et al. have put in considerable effort addressing the reviewers' comments. I am pleased with the edits and additions the authors have made, especially to the discussion sections, and believe this manuscript delivers compelling and interesting research.

- Thank you very much for your remarks! We are glad for your positive words on our revised manuscript and must convey our gratitude on your detailed comments and recommendations as they pointed us towards very helpful literature and greatly improved the content of our manuscript.